# A HIF independent oxygen-sensitive pathway for controlling cholesterol synthesis

Anna S. Dickson [1,10], Tekle Pauzaite[1,10], Esther Arnaiz[1,7,10], Brian M. Ortmann[1,8,10], James A. West[1], Norbert Volkmar[1,2,3], Anthony W. Martinelli [1], Zhaoqi Li[4,9], Niek Wit [1], Dennis Vitkup[5,6], Arthur Kaser [1], Paul J. Lehner [1] & James A. Nathan [1] ✉

Cholesterol biosynthesis is a highly regulated, oxygen-dependent pathway, vital for cell membrane integrity and growth. In fungi, the dependency on oxygen for sterol production has resulted in a shared transcriptional response, resembling prolyl hydroxylation of Hypoxia Inducible Factors (HIFs) in metazoans. Whether an analogous metazoan pathway exists is unknown. Here, we identify Sterol Regulatory Element Binding Protein 2 (SREBP2), the key transcription factor driving sterol production in mammals, as an oxygen-sensitive regulator of cholesterol synthesis. SREBP2 degradation in hypoxia overrides the normal sterol-sensing response, and is HIF independent. We identify MARCHF6, through its NADPH-mediated activation in hypoxia, as the main ubiquitin ligase controlling SREBP2 stability. Hypoxia-mediated degradation of SREBP2 protects cells from statin-induced cell death by forcing cells to rely on exogenous cholesterol uptake, explaining why many solid organ tumours become auxotrophic for cholesterol. Our findings therefore uncover an oxygen-sensitive pathway for governing cholesterol synthesis through regulated SREBP2-dependent protein degradation.

Cholesterol is an integral component of cell membranes, and is required for maintaining membrane fluidity and permeability. The cholesterol synthetic pathway also provides precursors for other metabolic pathways, including the production of steroid hormones, bile acids, and vitamin D[1]. Excess circulating cholesterol can lead to lipid accumulation and is a major risk factor for ischaemic heart disease[2]. Therefore, while cells require cholesterol to grow and divide, they must also control cholesterol abundance, which is achieved through a combination of regulated cholesterol uptake and synthesis.

Cellular cholesterol derives from two main sources: Dietary cholesterol circulates in high- and low-density lipoproteins (HDL and LDL), and can be directly taken up by cells for processing in the lysosome[3]. Alternatively, cholesterol can be synthesised within cells from acetyl-CoA. This occurs within the endoplasmic reticulum (ER) membrane, and involves a series of enzymatic steps governed by activation of the sterol response element binding protein 2 (SREBP2) transcription factor, and stabilisation of 3-Hydroxy-3-Methylglutaryl-CoA Reductase (HMGCR), the rate-limiting enzyme in cholesterol synthesis[4,5].

[1]Cambridge Institute of Therapeutic Immunology & Infectious Disease (CITIID), Jeffrey Cheah Biomedical Centre, Department of Medicine, University of Cambridge, Cambridge CB2 0AW, UK. [2]Institute for Molecular Systems Biology (IMSB), ETH Zürich, Zürich, Switzerland. [3]DISCO Pharmaceuticals Swiss GmbH, ETH Zürich, Zürich, Switzerland. [4]Department of Biology, Massachusetts Institute of Technology, Cambridge, MA, USA. [5]Department of Systems Biology, Columbia University, New York, NY, USA. [6]Department of Biomedical Informatics, Columbia University, New York, NY, USA. [7]Present address: Ochre-Bio Ltd, Hayakawa Building, Oxford Science Park, Edmund Halley Road, Oxford OX4 4GB, UK. [8]Present address: Biosciences Institute, Newcastle University, Herschel Building, Level 6, Brewery Lane, Newcastle upon Tyne NE1 7RU, UK. [9]Present address: Tango Therapeutics, 201 Brookline Ave Suite 901, Boston, MA, USA. [10]These authors contributed equally: Anna S. Dickson, Tekle Pauzaite, Esther Arnaiz, Brian M. Ortmann. ✉e-mail: jan33@cam.ac.uk

When cholesterol levels are high, SREBP2 is held in the ER through a sterol-sensitive interaction with SREBP cleavage-activating protein (SCAP) and insulin-induced genes (INSIG) 1 and 2[6]. This limits cholesterol synthesis by preventing SREBP2 trafficking to the Golgi apparatus and the subsequent processing that releases the N-terminal SREBP2 transcription factor[7]. Decreased cholesterol levels within the ER membrane liberate SREBP2 from SCAP/INSIGs, allowing SREBP2 to traffic to the Golgi and undergo intramembrane cleavage by Site-1 and Site-2 proteases[7]. The cleaved soluble N-terminal fragment forms a transcription factor complex, enters the nucleus and binds sterol responsive elements (SREs) to promote transcription of *HMGCR* and other cholesterol synthetic genes[8,9]. Alongside this transcriptional regulation of cholesterol synthesis, HMGCR levels are post-translationally controlled in a sterol-dependent manner by ubiquitination and ER-associated degradation (ERAD)[10–16]. Therefore, the combined sterol-sensitive processing of SREBP2 and ubiquitin-mediated regulation of HMGCR degradation provides a co-ordinated transcriptional and post-translational response to cholesterol abundance.

While all cells have the capacity to synthesise cholesterol, it is highly oxygen consuming, with 11 molecules of oxygen required to generate one molecule of cholesterol. Each oxygen-dependent step requires reduced nicotinamide adenine dinucleotide phosphate (NADPH) as a cofactor[17], and therefore oxygen availability and cellular redox impact on cholesterol production. In support of this, in fission yeast and other fungi, oxygen and sterol-sensing have co-evolved to share a common transcriptional programme that is governed by Sre1, the orthologue of SREBP2[18,19]. Fungal sterol pathways sense and respond to oxygen availability via prolyl hydroxylation, mediated by the oxygen-dependent prolyl hydroxylase Ofd1, with subsequent degradation of the cleaved N-terminal transcriptionally active portion of Sre1 (Sre1N)[18,20,21] in a mechanism strikingly similar to the oxygen-dependent prolyl hydroxylation of Hypoxia Inducible Factors (HIFs) in metazoans[22,23]. The mammalian orthologues of Ofd1 are not involved in sterol synthesis[24] and it has therefore been assumed that oxygen and sterol sensing pathways have diverged in metazoans, with SREBP2 providing transcriptional cholesterol homoeostatic control, whereas HIFs provide the oxygen-sensitive arm[25–27]. However, this does not explain frequent observations in cancers, where tumours adapt to their, often hypoxic, microenvironment by shutting down cholesterol synthesis to function as cholesterol auxotrophs[28,29]. Moreover, how SREBP2 function is controlled in hypoxia is not known.

Here, we uncover a mammalian SREBP2-mediated response for hypoxia-regulated cholesterol synthesis. This pathway is dominant over the canonical sterol sensing response, and shuts down cholesterol synthesis in a graded response to oxygen availability. Remarkably and unlike the oxygen-mediated regulation of HIFs or Sre1 in fungi, this pathway is independent of prolyl hydroxylation. Instead, it relies on a distinct mechanism for oxygen-sensing via the ER-resident ubiquitin ligase MARCHF6 and its redox responsiveness through activation by NADPH. Hypoxia promotes a relative accumulation of NADPH, which acts as a ligand for MARCHF6, increasing its activity[30] and promoting SREBP2 degradation. This oxygen and NADPH-sensitive response is co-ordinated with the HIF pathway to shutdown cholesterol synthesis and switch to cholesterol uptake when oxygen is scarce. Therefore, metazoans have evolved a distinct oxygen-sensitive transcriptional response to adapt to sterol availability, which underpins the cholesterol auxotrophy observed in hypoxic tumour microenvironments.

## Results

### Hypoxia promotes SREBP2 degradation
We first asked if SREBP2 processing or stability (Fig. 1a) was altered when oxygen is limiting, as has been observed for the fission yeast orthologue, Sre1[18]. Sterol deplete conditions were initiated by incubating cells in DMEM supplemented with lipid-depleted fetal calf

serum (FCS) and 10 µM mevastatin for 24 h, as previously described[12]. Cells were then exposed to 1% oxygen for an additional 16 h, and SREBP2 levels and processing measured by immunoblot (Fig. 1b, c). Sterol depletion resulted in the expected proteolysis of SREBP2 and accumulation of the N-terminal transcription factor (N-SRE) (Fig. 1b). However, incubation in 1% oxygen led to a marked decrease in full length SREBP2, and N-SRE failed to accumulate following sterol depletion (Fig. 1b, c). This hypoxia-mediated decrease in SREBP2 in HeLa cells was the direct opposite of the hypoxic regulation of fungal Sre1N, which is stabilised when oxygen availability is reduced[18]. However, similar findings were observed in several other cell lines, and were specific to SREBP2 (Supplementary Fig. 1a, b), as levels of the closely related SREBP1 did not alter in hypoxia (Supplementary Fig. 1c, d). Therefore, metazoans regulate SREBP2 abundance in hypoxia, but this is a distinct pathway to the hypoxic activation Sre1N in fission yeast.

The suppression of SREBP2 was not due to transcriptional downregulation (Fig. 1d), but resulted from increased protein degradation in hypoxia, as it was rescued by proteasome inhibition (20 nM bortezomib) (Fig. 1e). To determine the dynamics of this hypoxic-mediated degradation, we generated a fluorescently tagged SREBP2 knock-in reporter clone (SREBP2_Clover) in HeLa cells (Fig. 1f, Supplementary Fig. 1e), which was processed similarly to endogenous SREBP2 (Fig. 1f; Supplementary Fig. 1f). SREBP2-Clover levels started to reduce within 2 h of incubation in 1% oxygen, plateaued after 16 to 24 h (Fig. 1g; Supplementary Fig. 1g), and were again rescued by proteasome inhibition (Fig. 1h). Moreover, SREBP2_Clover degradation correlated with oxygen availability (Fig. 1i), confirming that SREBP2 is degraded rapidly in hypoxia, and shows a graded response to oxygen availability.

### Hypoxia overrides the canonical sterol-sensing response
The hypoxia-mediated degradation of SREBP2 degradation implied that oxygen availability not only regulates cholesterol synthesis but was dominant over the canonical sterol-sensing response. Decreased transcriptional activation of SREBP2 target genes (*HMGCR* and *HMGCS1*) (Supplementary Fig. 2a–d) in HeLa, HepG2 and HEK293T cells confirmed suppression of the transcriptional arm of cholesterol synthesis following sterol depletion in hypoxia. However, cholesterol production is mainly regulated by HMGCR stability. We therefore utilised an HMGCR knock-in reporter (HMGCR-Clover) cell line, used to define the degradation machinery for HMGCR processing in the ER membrane[12], to determine whether hypoxia governs cholesterol production.

Sterol depletion stabilised HMGCR-Clover as expected, but the combined exposure to 1% oxygen and sterol depletion suppressed HMGCR-Clover stabilisation (Fig. 2a), similarly to the reduction observed in HMGCR-Clover stablisation in mixed SREBP2 null cells (Supplementary Fig. 2e). We also observed that HMGCR-Clover levels were reduced in hypoxia following lipid depletion with a squalene mono-oxygenase (SQLE) inhibitor, NB-598 maleate, which inhibits cholesterol synthesis without directly binding HMGCR (Supplementary Fig. 3a). We next tested whether endogenous HMGCR levels were reduced, and observed that combined hypoxia and sterol depletion prevented HMGCR accumulation in HeLa, HepG2 and 293T cells (Fig. 2b; Supplementary Fig. 3b). HMGCR levels were rescued by proteasome inhibition (Fig. 2c, d), confirming that increased protein degradation in hypoxia was involved. Moreover, hypoxia prevented HMGCR-Clover accumulation irrespective of whether hypoxia coincided with the onset of sterol depletion, or occurred after 24 h (Fig. 2e), showing that hypoxia was dominant over the normal sterol-sensing response.

To directly measure the effect of hypoxia on cholesterol synthesis, we established [$^{13}$C]glucose liquid-chromatography mass spectrometry (LC-MS) tracing to track the incorporation of carbon units

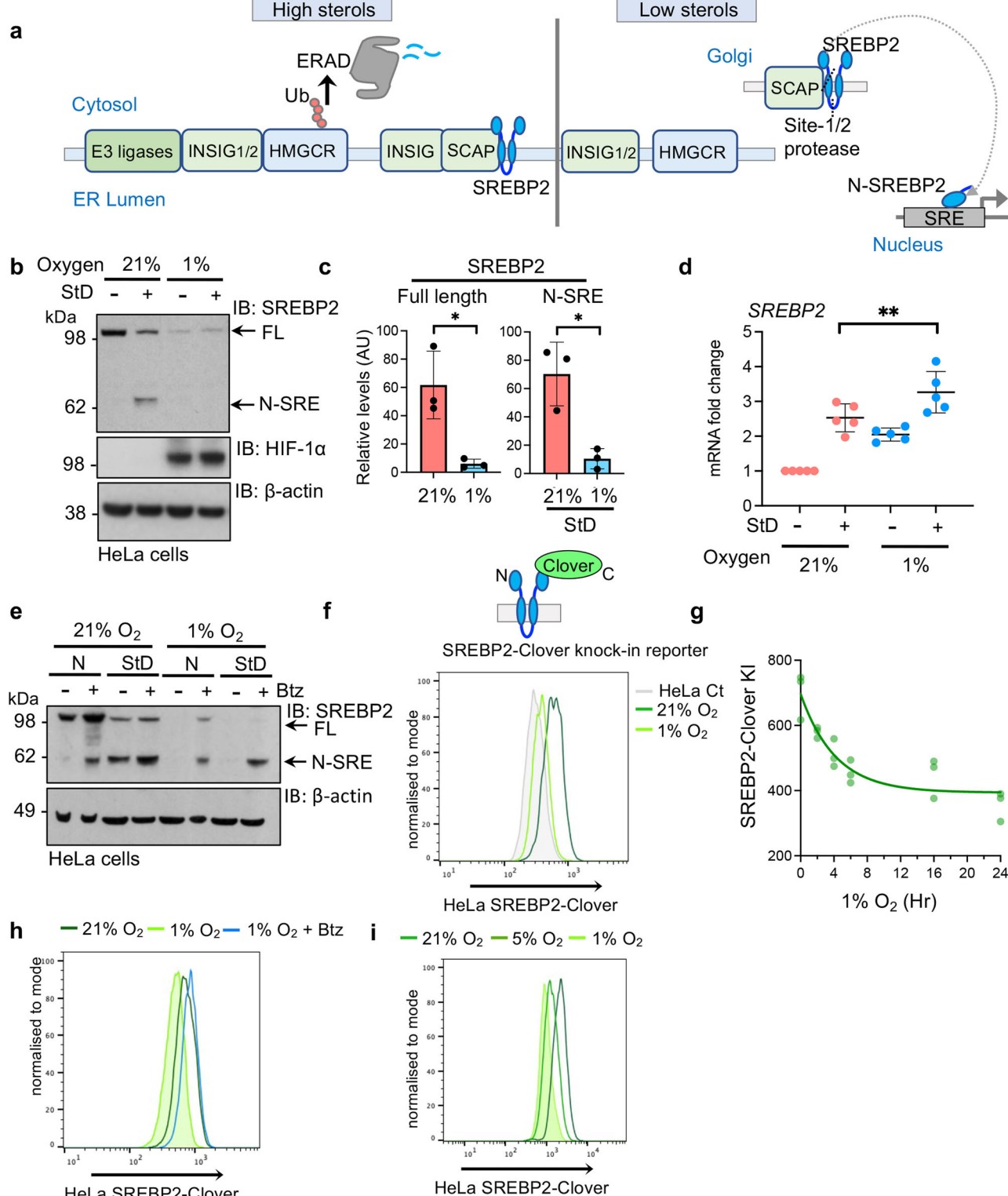

**Fig. 1 | Hypoxia promotes SREBP2 degradation. a** Schematic of known sterol-sensing pathway for regulating cholesterol synthesis at the ER membrane. **b**, **c** SREBP2 processing in HeLa cells treated with or without sterol depletion (StD: DMEM supplemented with 10% lipid-depleted FCS and 10 µM mevastatin) in 21% and 1% oxygen for 40 h. SREBP2 processing to release N-terminal transcription factor (N-SRE) determined by immunoblot (**b**) and quantified using ImageJ. Full length SREBP2 21% versus 1%, $P = 0.02$, N-SRE 21% versus 1% $P = 0.001$ (**c**). $n = 3$ biological repeats, mean ± SD. *$P \leq 0.05$ Two-way ANOVA. **d** SREBP2 mRNA levels with or without StD in 21% and 1% oxygen for 40 h. SREBP2 StD 21% versus 1%,

$P = 0.008$. $n = 5$ biological repeats, mean ± SD. **$P \leq 0.01$ Two-way ANOVA. **e** SREBP2 processing in normal media (N) or StD in 21% and 1% oxygen for 0 h, with or without addition of the proteasome inhibitor for 16 h prior to lysis (20 nM bortezomib, Btz). Immunoblot representative of 3 independent experiments. Flow cytometry analysis of HeLa SREBP2-Clover cells in 21% or 1% oxygen after 24 h (**f**), following incubation in 0–24 h 1% oxygen (**g**), and with proteasome inhibition (20 nM Btz 16 h) (**h**), and following StD and incubation in 21%, 5% and 1% oxygen for 24 h (**i**). Source data are provided as a Source Data file. $n = 3$ biological repeats. FL full length SREBP2, AU arbitrary units.

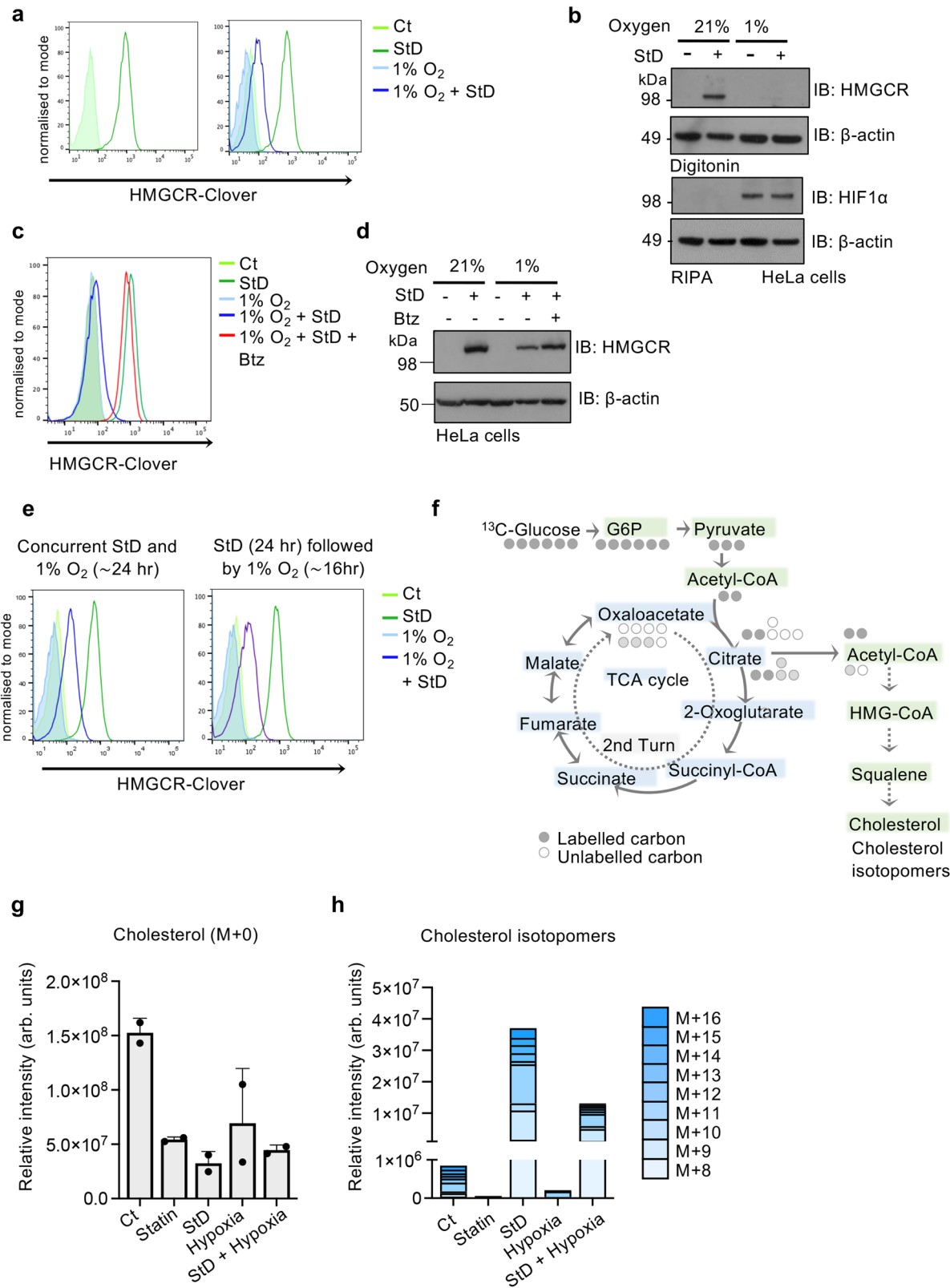

derived from glycolysis and acetyl-CoA into cholesterol isotopomers in HeLa cells (Fig. 2f; Supplementary Fig. 4). We validated this approach with statin treatment, which completely blocked cholesterol synthesis, with no [13C] molecules incorporated into cholesterol, despite the overall reduction in total cellular cholesterol content (Fig. 2g). In contrast, lipid depletion, which also reduced total cholesterol levels (Fig. 2g), caused a large increase in [13C]cholesterol isotopomers

(Fig. 2h; Supplementary Fig. 4). Hypoxia alone decreased [13C] incorporation into cholesterol molecules (Fig. 2g, h), confirming that the hypoxic suppression of SREBP2 prevented the transcriptional activation of cholesterol synthesis even when sterols were present in the media. Furthermore, this hypoxia-mediated suppression of cholesterol synthesis was dominant over the canonical sterol sensing response, with a marked reduction in [13C]cholesterol isotopomer formation

**Fig. 2 | Hypoxia overrides the canonical sterol-sensing response. a** Flow cytometry analysis of endogenous HMGCR-Clover levels following incubation in 21% or 1% oxygen, with or without StD for 40 h. **b** Endogenous HMGCR levels in HeLa cells following incubation in 21% or 1% oxygen, with or without StD, as described. Immunoblots representative of 3 independent experiments. HMGCR-Clover knock-in (**c**) or wildtype HeLa (**d**) were treated with StD or hypoxia as described, but also with the proteasome inhibitor bortezomib (20 nM for final 18 h). Representative of 4 independent experiments. **e** Concurrent treatment of HMGCR-Clover cells with StD and either 21% or 1% oxygen for 24 h (left); or pre-treatment of HMGCR-Clover HeLa cells with StD for 24 h followed by incubation in 21% or 1% oxygen for 16 h

(right). **f** Schematic of LC-MS analysis of [$^{13}$C]glucose uptake to trace incorporation of $^{13}$C into newly synthesised cholesterol. **g, h** Cholesterol isotopomers in HeLa cells cultured in control or lipid-depleted media and incubated in 21% or 1% oxygen for 24 h. HeLa cells were then supplemented with [$^{13}$C]glucose media for a further 24 h in the same conditions. Cells were also cultured in normal [$^{13}$C]glucose DMEM media with or without a statin treatment to control for the detection of newly synthesised cholesterol isotopomers. Cholesterol (M + 0) levels (**g**) and the most abundant isotopomers are shown (**h**), relative to cell counts. Source data are provided as a Source Data file. $P \leq 0.0001$. $n = 2$ biological repeats, mean ± SD. ***$P \leq 0.001$ Two-way ANOVA. Arb. units arbitrary units.

following combined hypoxia and lipid depletion (Fig. 2g, h). Therefore, hypoxic control of SREBP2 and HMGCR overrides the normal sterol synthetic response, and suppresses cholesterol production when oxygen availability is reduced.

## Oxygen-mediated regulation of cholesterol synthesis is independent of HIFs

HIFs provide the major transcriptional response to oxygen availability, and have been linked to HMGCR degradation in hypoxia through HIF-1 mediated transcription of *INSIG*[25]. We observed this HIF-mediated reduction in HMGCR stabilisation with either the 2-oxoglutarate dependent dehydrogenase inhibitor, DMOG, or the selective prolyl hydroxylase (PHD) inhibitor, Roxadustat (Fig. 3a; Supplementary Fig. 3c, d). However, HIF activation did not reduce HMGCR levels to the same extent as 1% oxygen (Fig. 3a; Supplementary Fig. 3c, d), indicating that a HIF-independent pathway was involved. To fully distinguish between HIF dependent and independent regulation, we abolished HIF signalling by deleting HIF1β (*ARNT*) with sgRNA (Supplementary Fig. 3e–g), which completely blocked induction of the HIF-1 target gene, Carbonic Anhydrase 9 (*CA9*) (Supplementary Fig. 3f, g), and measured the effect of hypoxia or PHD inhibition on SREBP2 or HMGCR levels.

HIF1β loss blocked the HIF mediated induction of *INSIG2* (Supplementary Fig. 3h), and prevented the increased degradation of HMGCR observed with PHD inhibition (DMOG) when cells were incubated in 21% oxygen and sterol depleted (Fig. 3b; Supplementary Fig. 3i). However, when HIF1β deficient cells were incubated in 1% oxygen and sterol depleted, HMGCR levels were suppressed to the same level as control HeLa cells (Fig. 3b; Supplementary Fig. 3i), and the hypoxia-mediated degradation of SREBP2 and its target genes was still observed (Fig. 3c–e). Therefore, the hypoxia-mediated regulation of SREBP2 is independent of the HIF pathway, and dominant over HIF-mediated *INSIG2* regulation.

We also demonstrated that the reduction in SREBP2 and HMGCR was proportional to oxygen availability, irrespective of the HIF pathway. Both SREBP2 and HMGCR levels were partially reduced in 5% oxygen, and nearly completely absent in 1% oxygen in both control and HIF1β deficient cells (Fig. 3f–h; Supplementary Fig. 3j). Furthermore, decreased SREBP2 levels correlated with decreased transcription of *HMGCR* in a HIF-independent manner (Supplementary Fig. 3k). Thus, hypoxic regulation of cholesterol synthesis is graded, relative to oxygen availability, and independent of the HIF response.

## MARCHF6 degrades SREBP2 in hypoxia

To find the degradative machinery which regulates SREBP2 and HMGCR in hypoxia, we undertook CRISPR/Cas9 genome-wide mutagenesis screens. We first identified genes that, when mutated, increased SREBP2-Clover levels, by iterative fluorescence activated cell sorting (FACS) (Fig. 4a). The ER-resident E3 ligase *MARCHF6* was the top hit (Fig. 4a). In a complementary approach we screened for genes that when knocked-out, increased HMGCR-Clover levels following sterol depletion in hypoxia (Fig. 4b). Again, *MARCHF6* was identified as a top hit, alongside known components

for HMGCR ERAD (*UBE2G2*, *AUP1* and *FAF2*) (Fig. 4b)[12]. SiRNA-mediated depletion of MARCHF6 in HeLa and SREBP-Clover reporter cells confirmed MARCHF6's involvement in SREBP2 degradation (Fig. 4d, e; Supplementary Fig. 5a). We also compared the effect of MARCHF6 and the TRC8 E3 ligase on SREBP2 levels as TRC8, although not identified in our screens, is an additional ER-resident ubiquitin ligase involved in SREBP2 turnover[31,32]. TRC8 depletion slightly increased SREBP2 levels, with MARCHF6 depletion being dominant, and their combined depletion synergistic (Fig. 4c, d; Supplementary Fig. 4a). Similar findings were observed with single MARCHF6 or TRC8 null clones, and combined MARCHF6/TRC8 KO cells (HeLa mCherry-CL1 reporter single or combined TRC8/MARCHF6 KO)[31] (Fig. 4e; Supplementary Fig. 5b, c). Furthermore, combined TRC8/MARCHF6 KO cells showed reduced SREBP2 ubiquitination following proteasome inhibition with bortezomib compared to control HeLa cells (Fig. 4f).

Having confirmed the requirement of MARCHF6, as well as TRC8, for SREBP2 stability, we examined whether preventing SREBP2 ubiquitination and degradation in hypoxia was sufficient to restore SREBP2 processing and re-establish the canonical sterol-sensing response of HMGCR. Combined MARCHF6/TRC8 KO cells (HeLa mCherry-CL1 reporter TRC8/MARCHF6 KO)[31] not only restored SREBP2 levels and processing in hypoxia (Fig. 4e), but also were sufficient to restore HMGCR levels (Fig. 4g, h). Individual clonal knock-outs of MARCHF6 or TRC8 partially restored N-SRE and HMGCR levels in 1% oxygen, with MARCHF6 being the dominant E3 ligase (Supplementary Fig. 5b, c). The degradation machinery for HMGCR in sterol replete conditions was the same in both 21% and 1% oxygen as previously described[12] (Supplementary Fig. 5d–f). Therefore, preventing the MARCHF6/TRC8-mediated degradation of SREBP2 in hypoxia restores the canonical cholesterol biosynthetic response upstream of HMGCR.

## The hypoxia cholesterol response is mediated by MARCHF6 NADPH sensing

The ability of MARCHF6 and TRC8 E3 ligases to control SREBP2 stability in both 21% and 1% oxygen did not adequately explain the more rapid degradation of SREBP2 in hypoxia. Cycloheximide chase and SREBP2 ubiquitination assays showed that the rapid ubiquitination and degradation of SREBP2 in hypoxia (Fig. 5b, c), was rescued by combined MARCHF6 and TRC8 depletion (Fig. 5c), and could not be explained by transcriptional regulation of *MARCHF6* and *TRC8* (Supplementary Fig. 6a, b). Therefore, E3 ligase activity must be increased in hypoxia.

NADPH was recently shown to regulate MARCHF6 catalytic activity through direct binding to a region with its C-terminus (MARCHF6-regulatory region, MRR)[30]. As NADPH is required as a cofactor for every oxygen-dependent step within cholesterol synthesis (Fig. 5d), hypoxia may alter the NADPH/NADP+ ratio, leading to increased MARCHF6 activity. In support of this, we observed an increase in NADPH levels when HeLa cells were exposed to hypoxia, which was markedly increased when combined with lipid depletion (Fig. 5e), consistent with less NADPH use during hypoxic suppression of lipid synthesis[33]. To test if this relative accumulation in NADPH was

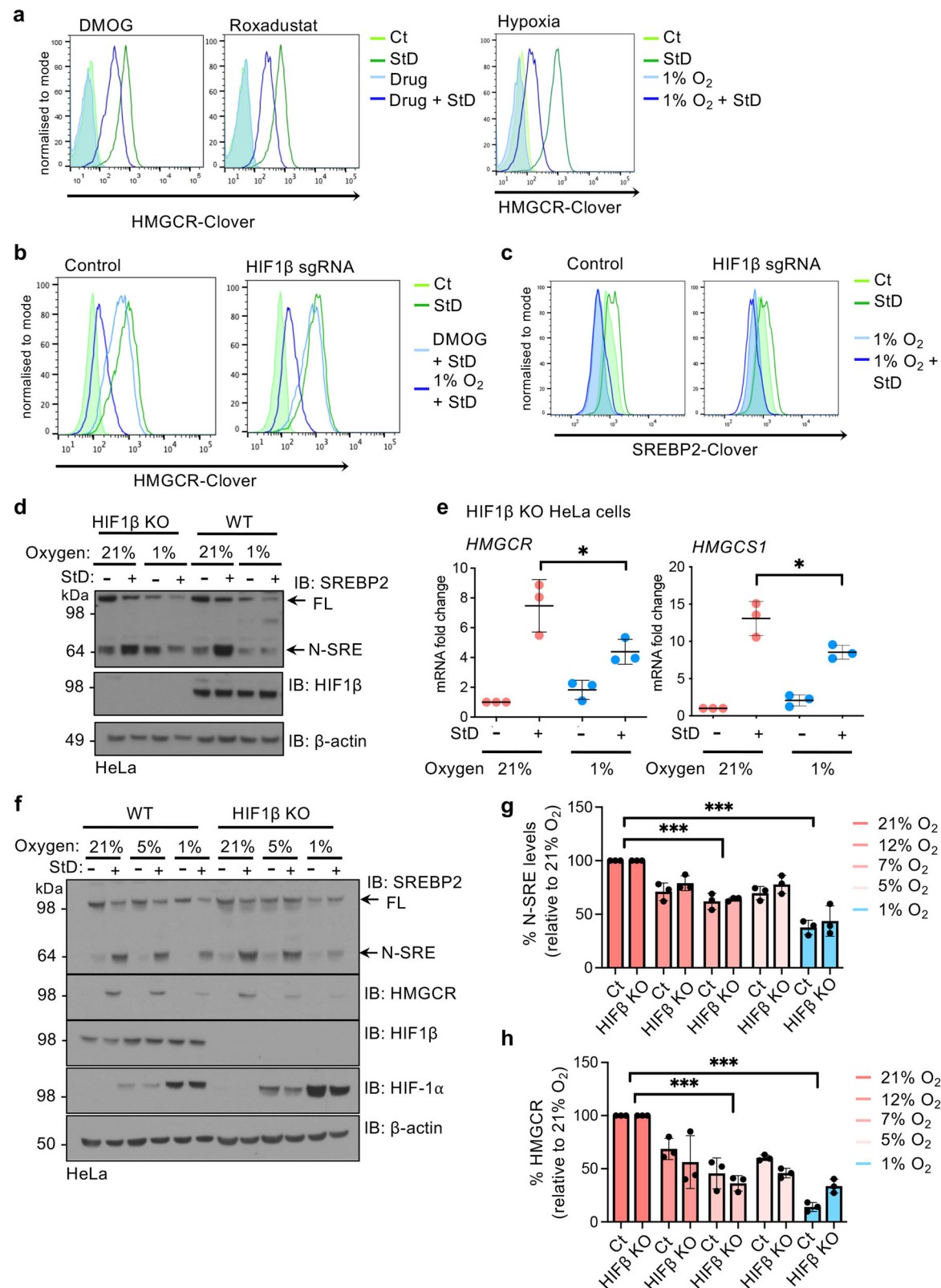

directly responsible for SREBP2 degradation, we depleted HeLa cells of NAD kinase (NADK) which reduces the total NADPH pool[30,34]. NADK depletion increased the levels of SREBP2 in 21% oxygen, and completely restored SREBP2 levels and processing of the N-terminal transcription factor in 1% oxygen (Fig. 5f, g), supporting the notion that NADPH acts as the ligand for MARCHF6 to mediate hypoxia-induced degradation of SREBP2.

## Hypoxia promotes cholesterol auxotrophy in tumour cells by suppressing cholesterol synthesis

Unlike fission yeast, where a single transcriptional pathway mediates oxygen and sterol sensing, we show that metazoans use two distinct oxygen-sensitive pathways to control cholesterol homoeostasis. MARCHF6 NADPH-mediated control of SREBP2 in hypoxia limits the oxygen consuming process of cholesterol synthesis, whereas the HIF

**Fig. 3 | Oxygen-mediated regulation of cholesterol synthesis is independent of HIFs. a** Flow cytometry analysis of HMGCR-Clover knock-in following StD and 1 mM DMOG or 100 μM Roxadustat (left panel), or following 1% oxygen (24 h) (right panel). **b** HMGCR levels in control or HIF1β null (mixed KO population) HMGCR-clover knock-in HeLa cells,with or without StD for 24 h, followed by incubation in 21% or 1% oxygen, or following 1 mM DMOG (16 h). **c** Flow cytometry analysis of SREBP2-Clover reporter levels in control HeLa or HIF1β null cells. Mixed KO populations of HIF1β were generated by sgRNA. Cells were cultured with or without StD for 40 h, and incubated in 21% or 1% oxygen for the final 16 h. Representative of 3 independent experiments. **d** Immunoblot of SREBP2 in HeLa HIF1β clonal KO cells compared to control HeLa (WT) following SD, with or without incubation in 1% oxygen. Representative of 3 independent experiments. **e** mRNA levels of the SREBP2 target genes *HMGCR* and *HMGCS1* in HIF1β clonal KO or control HeLa cells following StD, with or without incubation in 1% oxygen. StD 21% oxygen versus 1% oxygen *HMGCR*, $P = 0.011$, *HMGCS1*, $P = 0.05$. n = 3 biological repeats, mean ± SD. *$P ≤ 0.05$, Two-way ANOVA. **f–h** SREBP2 and HMGCR levels in HeLa control or Hela HIF1β KO cells cultured in 21%, 12%, 7%, 5% or 1% oxygen, with or without StD for 24 h. Immunoblot for proteins levels of SREBP2 and HMGCR for 21%, 5% and 1% (**f**). Protein levels in 12% and 7% oxygen shown in Supplementary Fig. 3j. Protein levels of N-SRE (**g**) and HMGCR (**h**) following StD were quantified by ImageJ following normalisation to β-actin. N-SRE Ct 21% versus 1% oxygen, $P < 0.0001$, N-SRE Ct 21% versus 1%, $P < 0.0001$, N-SRE HIF1β KO 21% versus 1% oxygen, $P < 0.0001$, N-SRE HIF1β KO 21% versus 1%, $P < 0.0001$, HMGCR Ct 21% versus 1% oxygen, $P < 0.0001$, HMGCR Ct 21% versus 1%, $P < 0.0001$, HMGCR HIF1β KO 21% versus 1% oxygen, $P < 0.0001$, HMGCR HIF1β KO 21% versus 1%, $P < 0.0001$. Source data are provided as a Source Data file. n = 3 biological repeats, mean ± SD. **$P ≤ 0.01$, ***$P ≤ 0.001$ Two-way ANOVA.

pathway promotes cholesterol uptake[26,29]. To understand how these pathways are co-ordinated, we examined the dependency of HeLa cells on cholesterol synthesis versus uptake by measuring their survival following statin treatment.

Simvastatin induced HeLa cell death after 48 h, with an $LD_{50}$ of ~5 μM simvastatin in 21% oxygen (Fig. 6a). However, cells conditioned to hypoxia by incubation in 1% oxygen for 48 h switched away from a reliance on cholesterol synthesis, and became more resistant to statins ($LD_{50}$ of ~10 μM) (Fig. 6a). HIF-driven cholesterol uptake was involved in this adaptive response, as HIF1β null cells failed to upregulate cholesterol uptake via Scavenger Receptor Class B Member 1 (*SCARB1*) transcription[27,29], and died similarly in 1% and 21% oxygen (Supplementary Fig. 7a, b). However, hypoxia-mediated degradation of SREBP2 was equally important, as combined MARCHF6/TRC8 depletion prevented the increased resistance to simvastatin in hypoxia, and identical cell death was observed in both 21% and 1% oxygen (Fig. 6b). Therefore, to distinguish which pathway was dominant, we overexpressed SREBP2 and asked if it was sufficient to restore the cholesterol synthetic pathway and sensitivity to statins (Fig. 6c–e). SREBP2 overexpression not only restored SREBP2 levels and processing in hypoxia, but activated cholesterol production through HMGCR stabilisation (Fig. 6c–e). However, this restoration of the cholesterol synthetic pathway came at a high cost, with all cells showing a markedly increased sensitivity to statin-induced cell death, dying in ~ 5 μM simvastatin (Fig. 6f), confirming that SREBP2 is the key determinant of cholesterol synthesis in hypoxia.

Finally, we showed that hypoxic regulation of SREBP2 helps facilitate the switch to cholesterol auxotrophy, known to occur in the development of kidney cancer[29]. Clear cell renal cell carcinomas (ccRCCs) are cholesterol auxotrophs and die in lipid-deplete media (Fig. 6g), whereas renal tumour epithelial (RTE) cells (HK-2 cells), from which ccRCCs arise, rely on cholesterol synthesis (Fig. 6g)[29]. Both RTE (HK-2) and ccRCC (RCC4) cell lines showed a hypoxia-dependent decrease in SREBP2 levels (Supplementary Fig. 7c, d), confirming the conserved nature of the oxygen-sensitive SREBP2 response across cell types. Moreover, this hypoxia-mediated reduction in cholesterol synthesis prevented statin-induced death in HK-2 cells, with ~ 90% viability following 20 μM simvastatin in 1% oxygen, compared to 50% in 21% oxygen (Fig. 6h). Therefore, the hypoxia-mediated shut down of cholesterol synthesis facilitates the switch to cholesterol auxotrophy.

Together, our findings establish a role for a HIF-independent pathway for terminating cholesterol synthesis in hypoxia, and demonstrates how this response is co-ordinated with the HIF pathway to promote cell growth in low oxygen environments (Fig. 7).

## Discussion

The close relationship between oxygen availability and cholesterol synthesis is observed across species. In fission yeast, oxygen and sterol sensing pathways converged to co-ordinate Sre1 stability and activation of the Sre1 transcriptional response[20]. Here, we uncover a distinct mammalian oxygen-sensitive pathway for controlling cholesterol synthesis via degradation of SREBP2. This contrasts with Sre1, where hypoxia stabilises the transcription factor, and provides one of the first examples of hypoxia promoting protein degradation.

A striking outcome of our studies is the HIF-independent nature of the response. Prior reports placed HIF-1 as the oxygen-sensitive regulator of cholesterol synthesis, through decreased HMGCR degradation via HIF-1 dependent transcription of *INSIG2*, or via the accumulation of lanosterol in hypoxia[25,35]. These observations cannot explain our findings, as we show that both SREBP2 and HMGCR are degraded to the same extent, irrespective of HIF activity. Moreover, our use of lipid-depleted FCS combined with a statin prevents lanosterol formation, and as lanosterol is unable to bind SCAP-INSIGs[36], it cannot account for the hypoxic regulation of SREBP2. Therefore, while HIF activation is clearly required to support cholesterol uptake in hypoxia, HIFs only play a minor role in controlling HMGCR stability, with no involvement in the upstream, and predominant oxygen-mediated regulation of SREBP2.

MARCHF6 is the dominant ligase for hypoxia-mediated SREBP2 degradation, with TRC8, making a smaller contribution. MARCHF6 is also involved in the regulation of a second rate-limiting step in cholesterol synthesis, squalene mono-oxygenase[37]. Hypoxia increases the activity of squalene mono-oxygenase[38], promoting the latter part of cholesterol synthesis. However, our metabolic tracing analysis demonstrate that cholesterol synthesis is rapidly shut down in hypoxia, indicating that hypoxia-mediated degradation of SREBP2 is dominant in shutting down the synthetic pathway.

As hypoxia-specific recruitment of MARCHF6 could not adequately explain the rapid SREBP2 loss, we postulated that oxygen abundance would markedly alter the NADPH/NADP+ pools, impacting on MARCHF6 activity[30]. Global analysis of NADPH/NADP+ levels confirmed a relative accumulation of NADPH in hypoxia, which was further exacerbated by sterol depletion. Local changes in NADPH levels at the ER membrane may be even more profound in hypoxia, as all oxygen-dependent steps in the synthetic pathway use NADPH as a cofactor. Interestingly, this accumulation in NADPH occurs alongside a general deficiency in NAD+, which drives cancer cells away from fatty acid uptake to rely on lipid auxotrophy in hypoxia[33]. Our finding that NADK depletion prevents MARCHF6-mediated degradation of SREBP2, strongly implicates NADPH as the ligand facilitating the oxygen-sensitivity of the SREBP2 degradative response. Whether TRC8 is also regulated by NADPH remains to be determined, but TRC8 may control constitutive SREBP2 degradation, while MARCHF6 senses changes in oxygen/NADPH levels. It also remains possible that deubiquitinating enzymes (DUBs) may be regulated by oxygen or NADPH levels, although we have not yet identified a DUB involved in controlling SREBP2 ubiquitination.

We show that redox-sensitive ligands, such as NADPH, provide a dynamic oxygen-sensing response through regulated protein degradation. While entirely distinct from oxygen-sensing by PHDs within HIF

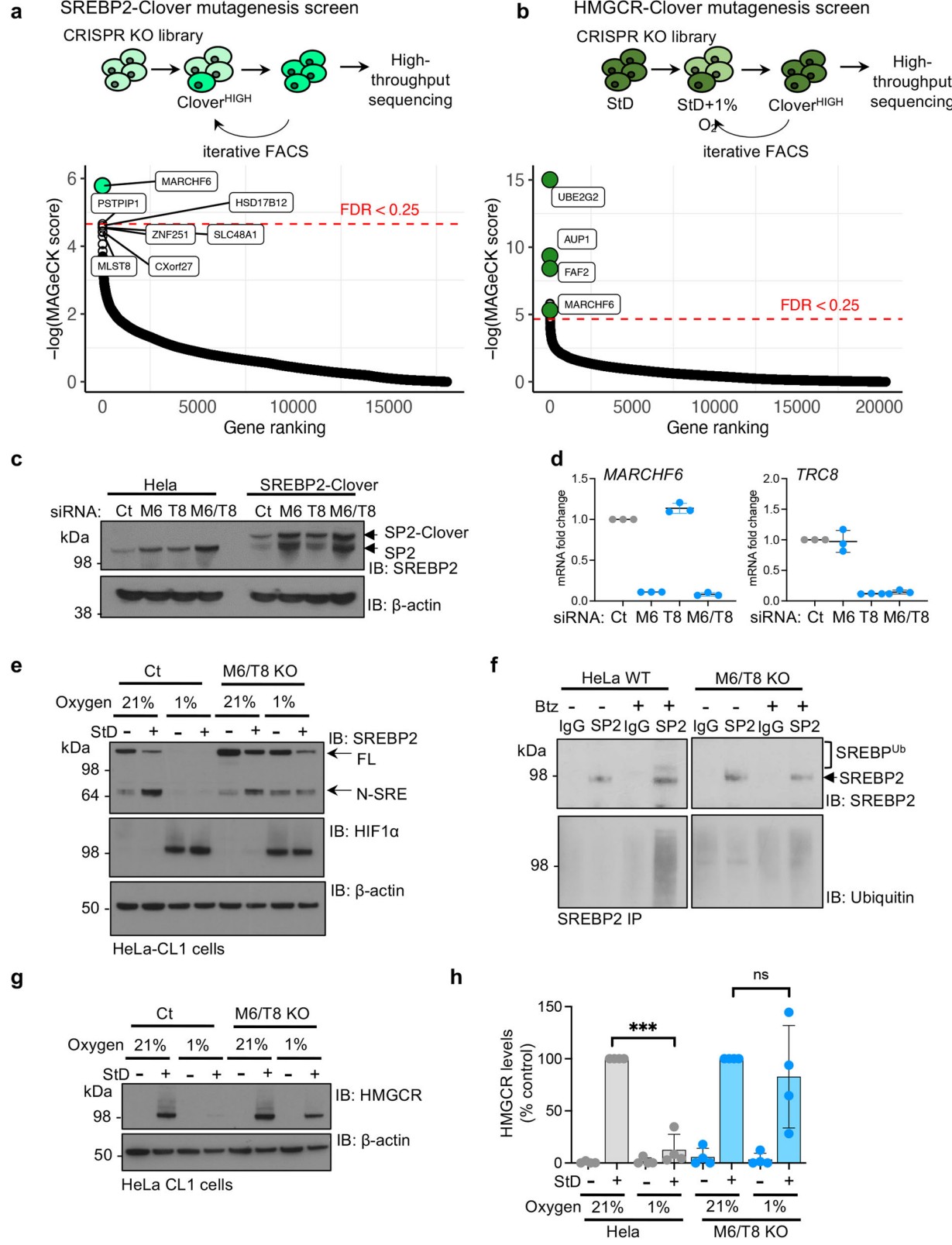

pathway, both responses share common features: (1) Oxygen availability correlates with enzyme activity of PHDs and MARCHF6, (2) both involve recruitment or activation of a ubiquitin ligase, and (3) they both regulate transcription factors. However, NADPH-mediated activation of MARCHF6 in hypoxia promotes protein degradation, whereas HIFs are stabilised in hypoxia. In addition, pVHL, the ligase controlling HIF stability is not itself regulated by oxygen, but is

recruited by HIF-prolyl hydroxylation. Therefore, the recruitment of redox-sensitive ligands for E3 ligase activity opens up new avenues for understanding how cells adapt to their oxygen environments.

Our studies show that both the HIF and SREBP2 transcriptional pathways have co-evolved to regulate cellular cholesterol metabolism in an oxygen-dependent manner. Why both pathways have diverged from a common pathway in fission yeast, likely relates to the

**Fig. 4 | MARCHF6 degrades SREBP2 in hypoxia. a, b** Summary of parallel CRISPR/Cas9 mutagenesis screens using the SREBP2-Clover reporter (**a**) or HMGCR-Clover reporter cells (**b**). **a** Mutagenised SREBP2-Clover cells were sorted for a Clover^HIGH population in 21% oxygen by FACS at day 9, and underwent a second sort at day 17 to enrich for this population. **b** Mutagenised HMGCR-Clover cells were sorted for a Clover^HIGH population in StD conditions (42 h) and incubated in 1% oxygen for the last 18 h (day 9). The enriched population underwent a second sort at day 14 to enrich for this Clover^HIGH population. SgRNA were identified by Illumina NovaSeq (**a**) or HiSeq (**b**), and compared to mutagenized population that had not undergone phenotypic selection. Comparative bubble plots are shown. Unadjusted *P* value calculated using MaGECK robust rank aggregation (RRA); red line false discovery rate (FDR) ≤ 0.25. Benjamini-Hochberg FDR. **c, d** Endogenous SREBP2 and SREBP2-Clover knock-in levels in HeLa cells following siRNA-mediated depletion of MARCHF6 (M6), TRC8 (T8), or combined M6/T8 depletion (**c**). mRNA levels of *MARCHF6*, *TRC8* and *SREBP2* confirm siRNA-mediated depletion of the E3 ligases

(**d**). *n* = 3 biological repeats, mean ± SD. **e** HeLa mCherry-CL1 control (Ct) or MARCHF6/TRC8 double knockout clonal cells were treated StD for 42 h, with or without 1% oxygen for the final 18 h. SREBP2 processing was analysed by immunoblot. Representative of 3 biological repeats. **f** Ubiquitination of SREBP2. Immunoprecipitation of SREBP2 in HeLa control or combined MARCHF6/TRC8 siRNA-depleted cells. Prior to lysis, cells were treated with or without Btz (5 μM 6 h). Immunoprecipitated SREBP2 was analysed by SDS–PAGE and immunoblotted for ubiquitin. Representative of 3 biological repeats. **g, h** HeLa mCherry-CL1 control (Ct) or MARCHF6/TRC8 double knockout clonal cells were treated StD for 42 h, with or without 1% oxygen for the final 18 h. HMGCR levels (**g**) were analysed by immunoblot. HMGCR levels were quantified using ImageJ (**h**). HMGCR StD 21% versus 1% oxygen, *P* < 0.0001, M6/T8 KO StD 21% versus 1% oxygen, *P* = 0.89. Source data are provided as a Source Data file. *n* = 4 biological repeats, mean ± SD. ***$P$ ≤ 0.001 Two-way ANOVA.

complexity of lipid homoeostasis in multicellular organisms, and the fluctuating oxygen and nutrient gradients within tissues. SREBP2 degradation in hypoxia must be fast, to shut down the oxygen consuming nature of cholesterol synthesis. HIF regulation of cholesterol uptake and HMGCR degradation is slower, as most HIF target genes take several hours to upregulate. However, HIF activation does seem to be required to promote cholesterol uptake as part of the adaptive response to shutting down cholesterol biosynthesis, and provides resistance to statins[29,39]. How hypoxia and HIFs co-ordinate cholesterol uptake remains to be fully determined. *SCARB1* is transcriptionally upregulated by HIFs, but the involvement of hypoxia or HIFs in controlling other aspects of cholesterol uptake, such as transcrtional regulation by Liver X Receptors (LXRs) and LDL receptor levels by Idol, have not been examined[40]. The requirement for both pathways may also relate to the highly oxygen consuming nature of lipid synthesis, and downstream effects on the cholesterol pathway that we have not yet examined. It is possible that both hypoxic-mediated degradation of SREBP2 and HIF activation may alter the cholesterol ester storage pool. HIF-2 activation of *PLIN2* indicates that there is an important regulatory arm of the hypoxia response on lipid droplets[41]. How hypoxia-induced degradation of SREBP2 impacts on cholesterol esters and lipid droplets remains to be determined. However, global approaches to understand how oxygen is utilised in cancer cells show that it supports lipid synthesis for cell growth[33]. It is therefore imperative that oxygen abundance is rapidly linked to biosynthesis, and oxygen/NADPH sensing by MARCHF6 allows this.

The shift to cholesterol auxotrophy in certain tumours offers a survival advantage in the hypoxic TME. We now show that hypoxia-mediated degradation of SREBP2 provides the first important step to shut down cholesterol synthesis in low oxygen environments. Strikingly, hypoxic conditioning of tumour cells resets the cholesterol synthetic pathway and increases resistance to statin-induced cell death. These findings help explain why cholesterol synthesis is blocked in ccRCCs, and their resistance to statins[29]. Our findings have wider implications, as statins may be ineffectual in tissues that experience low oxygen environments, as cholesterol synthesis is already suppressed. Moreover, cells that experience large physiological oxygen gradients, such as immune cells migrating through tissues, will be susceptible to the hypoxic control of cholesterol synthesis that may perturb their function.

## Methods

### Cell lines and reagents

HeLa, HEK293T, RCC4, HK-2, HepG2, cells were maintained in DMEM (Sigma Aldrich D6429) and supplemented with 10% foetal calf serum (FCS) and 100 units/ml penicillin with 100 μg/ml streptomycin, in a 5% $CO_2$ incubator at 37 °C. 786-O cells were cultured in RPMI-1640 (Sigma R8758). HeLa HMGCR-Clover cells were generated as previously described[12]. HeLa mCherry-CL1 cells have previously been described[31].

Cells were confirmed mycoplasma negative (Lonza, MycoAlert), and authenticated by short tandem repeat profiling (Eurofins Genomics). Cells were also cultured with lipid-depleted serum as described for lipid-deplete or sterol deplete conditions. Full details of reagents and antibodies used are shown in Supplementary Table 1.

### Plasmids

All plasmids are detailed in the Supplementary Table 2. CRISPR sgRNAs were cloned into a lentiviral sgRNA expression vector pKLV-U6gRNA(BbsI)-PGKpuro2ABFP46[42] or LentiCRISPRv2[43] as previously described[44]. Cas9 expressing cells were generated using LentiCas9 with puromycin, hygromycin or blasticidin as previously described[45]. sgRNA sequences are detailed in Supplementary Table 3.

SREBP2 constructs were generated from an I.M.A.G.E cDNA clone (Source Bioscience IRATp970B0781D 6169568) and cloned into the pHRSIN-pSFFV-HA backbone with puromycin resistance using NEBuilder HiFi (NEB). The primers are detailed in Supplementary Table 4.

### Lentiviral production and transduction

Lentivirus was produced by transfection of HEK293T cells (Trans-IT 293 reagent, Mirus) or Fugene (Promega) at 70–80% confluency in six-well plates, with the appropriate pHRSIN vector and the packaging vectors pCMVR8.91 (gag/pol) and pMD.G (VSVG), as previously described[44]. Viral supernatants were harvested at 48 h, filtered (0.45 μm filter), and stored at −80 °C. For transduction, cells were seeded on 24-well plates in 500 μl media, and 500 μl viral supernatant added. Plates were centrifuged at 1800 rpm (600 × *g*) at 37 °C for 1 h. Antibiotic selection was applied from 48 h.

### CRISPR−Cas9 targeted deletions

Gene-specific CRISPR sgRNA sequences were taken from the TKO library, designed using E-CRISP (http://www.e-crisp.org/E-CRISP/) or VBC score (https://www.vbc-score.org/), with 5′CACC and 3′CAAA overhangs, respectively. SgRNAs were ligated into the LentiCRISPRv2 or pKLV-U6gRNA(BbsI)-PGKpuro2ABFP vector and lentivirus produced as described. Transduced cells were selected with puromycin, and were generally cultured for 9–10 days before subsequent experiments to allow sufficient times for depletion of the target protein. KO clones were isolated from the sgRNA-targeted populations by serial dilution or FACS. SgRNAs used are shown in Supplementary Table 3.

### CRISPR/Cas9-mediated gene knock-in

The Hela HMGCR-Clover knock-in cells were generated as previously described[12]. HeLa SREBP2-Clover cells were generated using a knock-in donor template, in a similar method to HMGCR-Clover. SREBP2 homology arms were synthesised as gBlocks (IDT) and cloned into pDonor-Clover-LoxP using Gibson assembly (NEBuilder HiFi DNA), with -1 kb flanking homology arms An EcoRV site was inserted upstream of KpnI in pDonor-Clover-LoxP to allow homology arms to

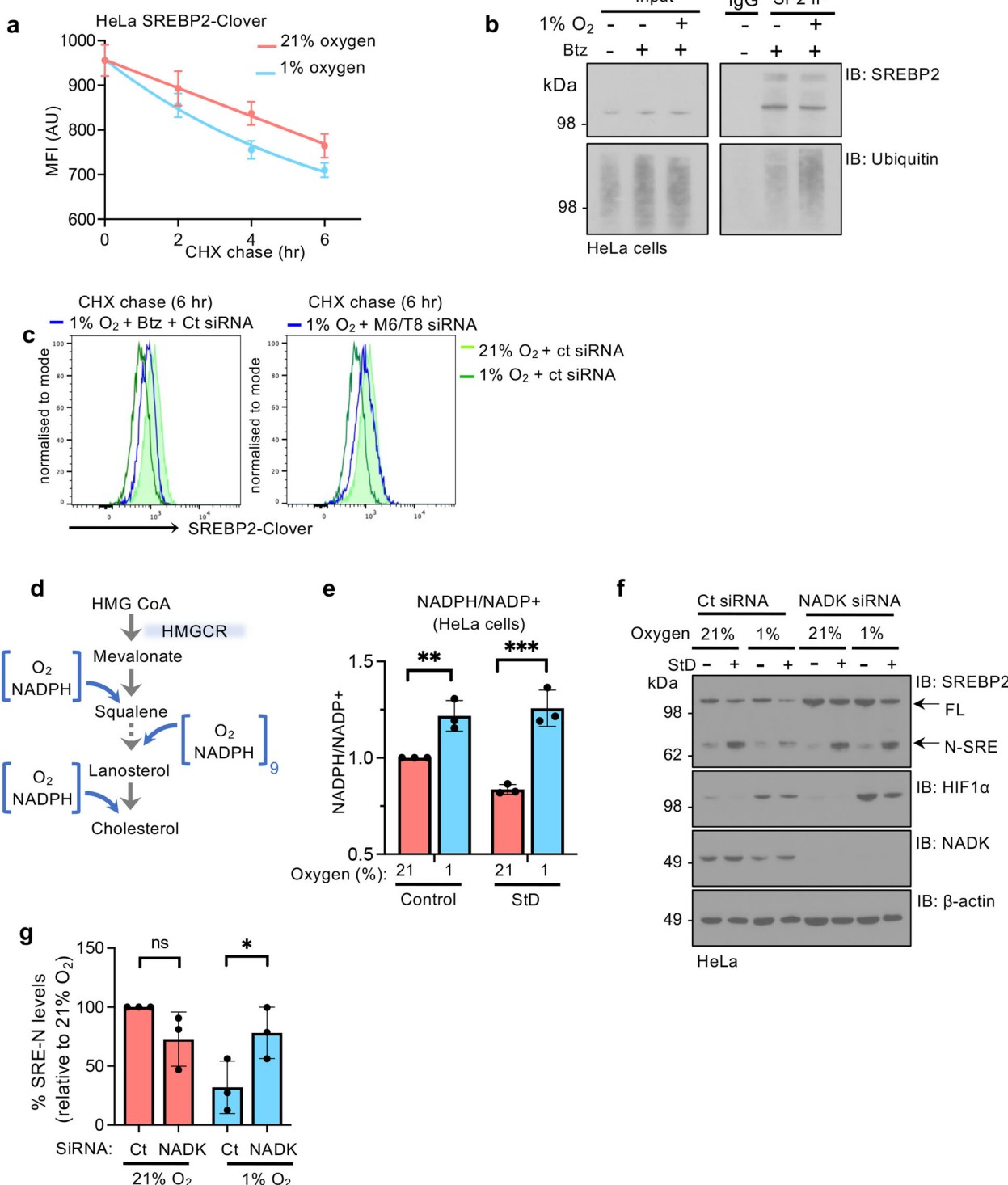

**Fig. 5 | The hypoxia cholesterol response is mediated by MARCHF6 NADPH sensing. a** SREBP2-Clover levels by flow cytometry in HeLa SREBP2-Clover knock-in cells treated with 10 μg/ml of cycloheximide for 0–6 h with or without treatment with 1% oxygen. Quantification of mean fluorescence intensity (MFI) shown. $n = 5$ independent biological replicates, mean ± SD. **b** Ubiquitination of SREBP2 in hypoxia. Immunoprecipitation of SREBP2 in HeLa cells incubated in 21% or 1% oxygen for 6 h, with or without Btz (5 μM 6 h). Immunoprecipitated SREBP2 was analysed by SDS–PAGE and immunoblotted for ubiquitin. Representative of 3 biological replicates. **c** SREBP2-Clover levels by flow cytometry in reporter cells following treatment with 10 μg/ml of cycloheximide for 6 h and 1% oxygen, with or without Btz (5 μM 6 h) (left panel), or following combined siRNA-mediated depletion of MARCHF6 (M6) and TRC8 (T8) (right panel). The data depicted in the left and right panels originated from the same experiment and as such the control plots are the same in both.

Representative of 3 biological replicates. **d** Schematic of oxygen consuming and NADPH oxidation steps within the cholesterol synthetic pathway. **e** Relative NADPH/NADP+ levels in HeLa cells following incubation in 21% or 1% oxygen, with or without StD for 24 h. Control 21% versus 1% oxygen, $P = 0.006$, StD 21% versus 1% oxygen, $P < 0.0001$. $n = 3$ biological repeats, mean ± SD. **$P ≤ 0.01$, ***$P ≤ 0.001$ Two-way ANOVA. **f**, **g** HeLa cells were depleted for NADK using siRNA or a mock control siRNA. After 48 h cells were incubated in 21% or 1% oxygen, with or without StD for 24 h. SREBP2 processing was visualised by immunoblot (**f**), and protein levels of N-SRE following sterol deplete conditions were quantified by ImageJ, following normalisation to β-actin (**g**). Control versus siRNA NADK in 21% oxygen, $P = 0.232$, Control versus siRNA NADK in 1% oxygen, $P = 0.038$. Source data are provided as a Source Data file. $n = 3$ biological repeats, mean ± SD. *$P ≤ 0.05$, Two-way ANOVA.

be inserted without altering the Clover-LoxP cassette (Supplementary Table 4). The vector was cut using EcoRI and PvuI for the 5′ homology arm, and PacI and EcoRV for the 3′ homology arm. The final vector, pDonor-Clover-LoxP-SREBP2, was sequence verified and transfected into HeLa cells alongside Cas9 and sgRNA targeting the 3′ SREBP2 untranslated region (pSpCas9(BB)-T2A-Puro vector) (Supplementary Table 4). Cells were treated with puromycin for 14 days and expanded. The presence of SREBP2-Clover was verified by flow cytometry and immunoblot. Cells were then transfected with pHRSIN- pSFFV Cre pGK Hygro to remove the puromycin resistance and single cell clones isolated. HeLa SREBP2-Clover KI was confirmed by PCR amplification using primers SPfor and SPrev 3′-GAGTGG-GAAGGAACAGGACAATTA-5′ (Supplementary Table 4). The presence of the 1596 bp fragment confirmed incorporation of Clover to the C-terminus in the knock-in cells (KI), and was not observed in the parental cells that only encoded WT SREBP2 (1269 bp).

## CRISPR−Cas9 forward genetic screens

HeLa HMGCR-Clover screen: Hela HMGCR-Clover cells were first transduced with Cas9-Hygro and maintained under selection for at least 14 days. For the screen, $10^8$ cells were transduced with the CRISPR knockout Bassik library[46]. Cells were transduced at an MOI of ~30% with the transduction efficiency determined by measurement of mCherry by flow cytometry 48 h post transduction. Transduced cells were enriched by puromycin selection for at least seven days. To maintain an even representation of guides throughout the screen the unsorted library was routinely maintained in excess of $100 \times 10^6$ cells ensuring approximately 500X representation of each guide. After eight days, cells were seeded at 30% confluency and allowed to settle before being sterol depleted (DMEM + 10% LPDS + 10 μM mevastatin+penicillin/streptomycin) overnight. The next day, the cells were placed in hypoxia for 18 h. Cells were prepared for FACS sorting, and Clover^HIGH cells were collected. These cells were cultured for a further six days before being subjected to sterol depletion and hypoxia and FACS as detailed above. DNA was extracted from the control library (transduced cells which had not undergone FACS) and sorted cells using the Puregene® Core kit A (Qiagen) according to manufacturer's instructions. The sgRNA locus was amplified using two rounds of PCR, and the amplicons sequenced by Illumina MiniSeq and HiSeq as previously described[47]. Reads were extracted, the first 19 bp trimmed using Cutadapt[48] (DOI:10.14806/ej.17.1.200) and then aligned against the Bassik sgRNA library using HISAT2[49]. Read counts for each sgRNA were compared between conditions, and false discovery rates for each gene were calculated using MAGeCK[50] (Supplementary Data 1). Amplification, indexing and sequencing primers are detailed in Supplementary Table 5.

HeLa SREBP2-Clover screen: The Toronto human knockout pooled library (TKOv3) was a gift from Jason Moffat (Addgene #125517)[51,52]. $7.2 \times 10^7$ HeLa SREBP2-Clover cells expressing Cas9 were transduced with pooled TKOv3 sgRNA virus to maintain greater than 200-fold coverage at multiplicity of infection 0.3. After 27 h cells were selected with puromycin 1 μg/ml, expanded and were pooled prior to any selection event to maintain representation throughout the screen. On day 9, $2.4 \times 10^8$ cells were harvested, washed in PBS supplemented with HEPES 10 mM, resuspended in sort media (PBS with 10 mM HEPES and 2% FBS) and filtered. FACS was undertaken using an Influx cell sorter (BD) selected the top and bottom 1% of fluorescent cells into collection media (50% DMEM, 50% FBS, 2% Penicillin-Streptomycin). Gating was set with reference to non-transduced controls treated with bortezomib or 1% oxygen. Selected cells were divided for expansion and for immediate DNA extraction (Puregene Core Kit A, Qiagen 51304, 158388). Simultaneously, $2 \times 10^7$ phenotypically non-selected transduced cells were harvested for library DNA extraction and $4 \times 10^7$ cells continued for the later library. Selected cells were further expanded before repeat sorting and DNA extraction on day 17. A two-stage PCR was completed to amplify inserts using the following primers, where

NNNNNNN represents a barcode for multiplexing (Supplementary Table 5) Following PCR, DNA was purified (AMPure XP, Agencourt A63880), quantified via Bioanalyzer (Agilent DNA 1000, 5067-1504) and sequencing performed by NovaSeq (custom primer, Supplementary Table 5). Reads were extracted and analysed with reference to the TKOv3 library as previously described (Supplementary Data 2).

## Sterol depletion

Cells were seeded at 60% confluency. The following day, cells were washed twice with PBS and before being incubated in sterol depletion media (DMEM, 10% LPDS, 10 μM mevastatin, penicillin/streptomycin). For hypoxic experiments, cells were sterol depleted for 24 h in 21% oxygen, prior to being incubated in 1% oxygen for a further 18 h still under sterol depletion. Where indicated, sterol depletion media was also added at the same time as the hypoxic incubation was started.

## Hypoxic incubation

All hypoxic experiments were performed in either a Whitley H35 Hypoxystation (Don Whitley Scientific) or a SCI-tive Dual Hypoxia workstation (Baker Ruskinn) maintained at 1–5% oxygen, 94% $N_2$, 5% $CO_2$ at 37 °C. During harvesting cells were kept upon ice to minimise re-oxygenation of samples

## Immunoblotting

Cells were lysed in an SDS lysis buffer (1% SDS, 50 mM Tris (pH 7.4), 150 mM NaCl, 10% glycerol and 5 μl ml−1 Benzonase (Sigma)) unless stated otherwise. Samples were incubated on ice for 10 min before heating at 90 °C for 5 min, 70 °C for 15 min (HMGCR or SREBP2). In specified cases cells were lysed using an appropriate volume of either RIPA buffer (50 mM TRIS pH 8.0, 150 mM NaCl, 0.1% SDS, 1% NP-40, 0.5% Sodium deoxycholate, protease inhibitors), or Digitonin buffer (1% digitonin, protease inhibitors). Cells were lysed on ice for at least 15 min before being subjected to centrifugation at $16.9 \times g$ for 10 min and the supernatants collected. 6× SDS sample buffer (Laemmli) was added to the supernatant, and the samples typically heated at 90 °C for 5 min prior to analysis. Proteins were separated by SDS−PAGE, transferred to PVDF (polyvinylidenedifluoride) membranes, probed with appropriate primary and secondary antibodies and developed using enhanced chemiluminescent or Supersignal West Pico Plus Chemiluminescent substrate (Thermo Scientific).

## Quantitative PCR

Total RNA was extracted using the RNeasy Plus minikit (Qiagen) following the manufacturer's instructions and then reversed transcribed using Protoscript II Reverse Transcriptase (NEB). Template cDNA (20 ng) was amplified using the ABI 7900HT Real-Time PCR system (Applied Biotechnology or Quantstudio 7, Thermo Scientific) reactions Transcript levels of genes were normalised to a reference index of a housekeeping gene (β-actin). The primers sequences are shown in Supplementary Table 6.

## Flow cytometry

Cells were harvested and washed twice with PBS by centrifugation before either being subjected to live cell flow cytometry (for Clover) or being fixed in 3.6% PFA in PBS. Cells were run on an LSRFortessa™ (BD Biosciences). Resulting data was analysed using the FlowJo software. For cell surface staining, cells were trypsinised, washed in PBS and incubated with 100 μl of the primary antibody made up in PBS for 30 min at 4 °C. The primary antibody was removed by a PBS wash by centrifugation ($500 \times g$, 5 min) before incubation with the appropriate Alexa Fluor conjugated secondary antibody for 30 minutes at 4 °C in the dark. Two final PBS washes were performed before cells were fixed in PBS with 3.7% PFA before analysis on a BD LSRFortessa™ (BD Biosciences).

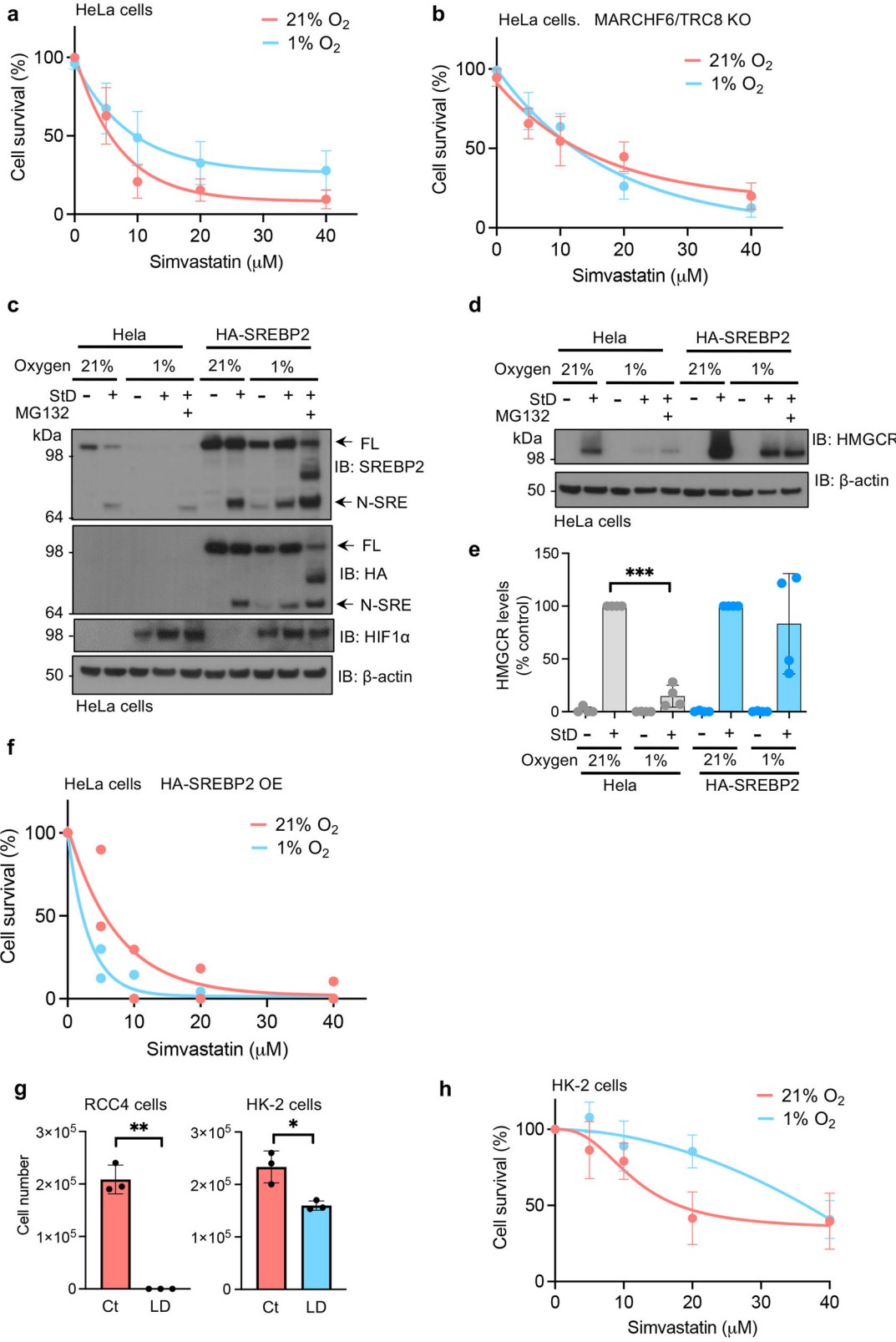

### siRNA-mediated depletion

HeLa cells were transfected with ON-TARGETplus siRNA (Dharmacon) for MARCHF6, TRC8 or NADK (Dharmacon). The MISSION siRNA Universal Negative Control (Sigma Aldrich) was used alongside, and siRNA were transfected uses Lipofectamine RNAi MAX (Thermo Fisher) according to manufacturer's instructions. Cells were harvested after 48 h for further analysis by flow cytometry, qPCR or immunoblot.

### Transient transfection assays

Cells were passaged 24 h prior to transient transfection to ensure they were in the exponential growth phase when transfected. Transient transfections in HeLa cells were carried out using TransIT-HeLaMonster® (Mirus) following the manufacturer's instructions. Briefly, the transfection mix (200 μl OptiMEM, 1 μg DNA, 5 μl TransIT and 1 μl Monster) was incubated for 30 min at

**Fig. 6 | Hypoxia promotes cholesterol auxotrophy in tumour cells by suppressing cholesterol synthesis.** HeLa cells (**a**), or HeLa MARCHF6 and TRC8 KO cells (**b**) were incubated in 21% or 1% oxygen for 48 h with simvastatin as indicated. Cell viability was measured by Hoechst staining at 48 h. $n = 7$ (**a**) and $n = 5$ (**b**) biological repeats, mean ± SD. **c** HeLa cells were transduced with HA-SREBP2 and StD for 42 h, with or without incubation in 1% oxygen as described. Additionally, cells were treated with 10 μM MG132 4 h prior to harvest where appropriate. SREBP2 levels were measured by immunoblot. Immunoblot representative of 2 biological repeats. **d, e** HeLa cells were transduced with HA-SREBP2 and StD for 42 h, with or without incubation in 1% oxygen as described. HMGCR levels were measured by immunoblot (**d**) and quantified using ImageJ (**e**). HMGCR StD 21%

versus 1% oxygen, $P < 0.0001$. $n = 4$ biological repeats, mean ± SD. ***$p \leq 0.001$ Two-way ANOVA. **f** HeLa cells transduced with HA-SREBP2 were incubated in 21% or 1% oxygen for 48 h with simvastatin as indicated. Cell viability was measured by Hoechst staining at 48 h. $n = 2$ biological repeats. **g** $10^5$ RCC4 renal cancer cells or HK-2 RTE cells were cultured in lipid-deplete media (LD) for 3 days and then the total number of viable cells was measured. RCC4 cells control versus LD media, $P = 0.006$, HK-2 cells control versus LD media 21%, $P = 0.04$. $n = 3$ biological repeats, mean ± SD. **$P \leq 0.05$, **$P \leq 0.001$ Unpaired two sample student T-test. **h** HK-2 cells were incubated in 21% or 1% oxygen for 48 h with simvastatin as indicated. Cell viability was measured by Hoechst staining at 48 h. Source data are provided as a Source Data file. $n = 3$ biological repeats, mean ± SD.

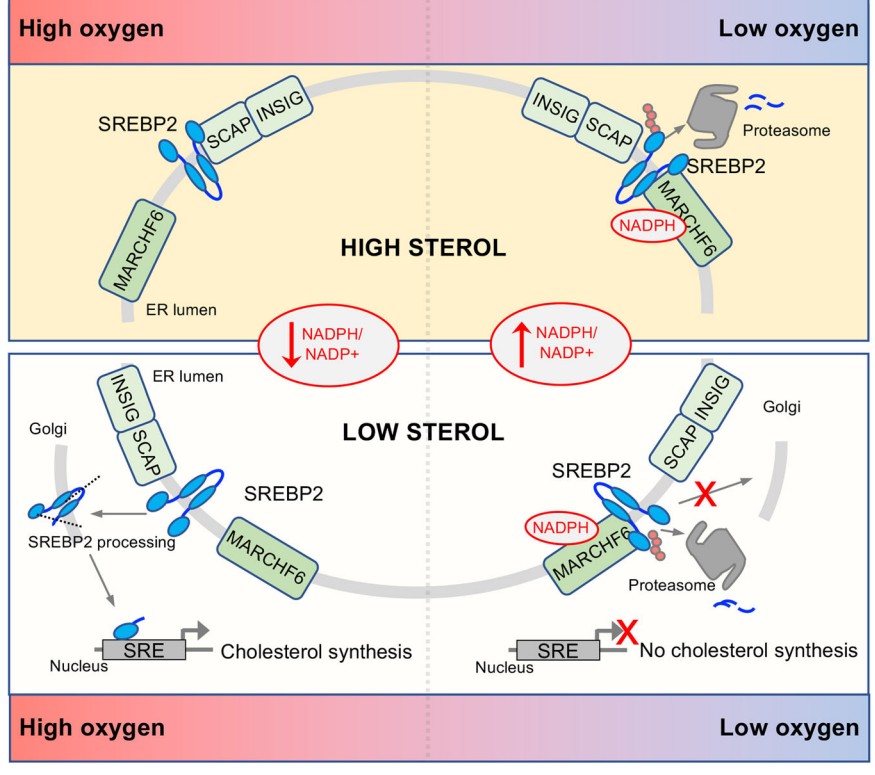

**Fig. 7 | Model of oxygen-dependent regulation of cholesterol biosynthesis.** Schematic of the oxygen-dependent regulation of SREBP2. In 21% oxygen and sterol replete conditions, SREBP2 is held in the ER through its interaction with SCAP/INSIGs, and basal SREBP2 turnover by MARCHF6 is low (top left). Sterol depletion releases SREBP2 from SCAP/INSIGs, and SREBP2 undergoes processing in the Golgi

to generate the N-SRE transcription factor (bottom left). In hypoxia, NAPDH accumulates, increasing MARCHF6 activity in a proportional manner to oxygen availability (top right). This promotes SREBP2 degradation in the ER, and this shuts down cholesterol synthesis in sterol deplete conditions (bottom right).

room temperature before being added dropwise to cells at 70% confluency in a 3 cm dish. 24 h post transfection, the cells were split 1:10 and the appropriate antibiotic selection added for up to 48 h to enrich for successful transfection.

### Cyclohexamide chase assays

HeLa SREBP2-Clover cells were seeded as previously described and sterol depleted for 24 h. Cells were incubated in 21% or 1% oxygen with 10 μg/ml of cyclohexamide and/or 5 μM bortezomib (Velcade) for 0, 2, 4, and 6 h and analysed by flow cytometry. Alternatively, cells were sterol depleted for 24 h, and then sterols were re-introduced to trigger HMGCR degradation (2 μg/ml 25-hydroxycholesterol and 20 μg/ml cholesterol). Cells were harvested at the indicated time points before analysis by immunoblot.

### SREBP2 Immunoprecipitation and ubiquitination assays

HeLa cells were seeded in 10 cm plates, and when they were at 70% confluency they were incubated in 1% oxygen with or without 5 μM bortezomib (Velcade) for 6 h. Cells were lysed in 500 μl of RIPA (25 mM Tris•HCl pH 7.6, 150 mM NaCl, 1% NP-40, 1% sodium deoxycholate, 0.1% SDS) supplemented with cOmplete Protease Inhibitor Cocktail and Denarase for 30 min at 4 °C. Lysates were centrifuged at 14,000 r.p.m. for 10 min, supernatants collected and then diluted with 500 μl of 1% Triton for preclearing with Protein G magnetic beads (Thermo Scientific) for 2 h at 4 °C. Supernatants were then incubated with primary antibody overnight (rotation at 4 °C). Protein G magnetic beads were then added for 2 h, and samples were then washed three times with 1% Triton and once with PBS. Bound proteins were eluted in 1× SDS loading buffer, separated by SDS–PAGE and immunoblotted.

## Statin and lipid depletion sensitivity assays

**Lipid depletion.** HK-2 and RCC4 cells were plated on six-well plates at a density of $5 \times 10^4$ for HK-2 or $1 \times 10^5$ for RCC4 cells and cultured in DMEM, with 10% FCS or 10% LPDS for 3 days before cells were counted using a hemocytometer.

**Statin sensitivity.** HK-2 or HeLa cells were plated on 96 well plates at a density of $5 \times 10^3$ cells per well for HeLa and $1 \times 10^4$ for HK-2. After 24 h, cells were treated with 0–40 μM of Simvastatin and/or 1% oxygen for 48 h. Cells were washed with PBS, stained with 100 μl 2.5 μM Hoesht in PBS, and visualised using CLARIOstar Plate Reader at 355–20/455–30 nm.

## NADPH/NADP+ analysis

$1 \times 10^6$ HeLa cells were cultured in RPMI with or without sterol depletion, and/or hypoxia (1% Oxygen) for 48 h before harvesting the cells. The NADP/NADPH Assay Kit (Abcam) was used to measure NADPH and NADP+ levels. Cells were washed 3 times with ice cold PBS and scraped cells with 800 μL extraction buffer, prepared as directed by the manufacturer's protocol, and analysed in a plate reader (CLARIOstar 5.60 R2) at OD450 nm.

## Liquid-chromatography metabolic measurements

[$^{13}$C]glucose labelling and isotope tracing. HeLa cells were seeded on 15 cm plates and allowed to recover. At 40% confluency the cells were cultured in DMEM or DMEM containing lipid-depleted FCS, or DMEM containing 5 μM mevastatin at 21% or 1% oxygen for 24 h. Cells were then supplemented with media containing [13 C]glucose and labelled for a further 24 h. Cells were washed in PBS and then harvested using trypsin. The cells were then washed twice in PBS and pellets were frozen at −80° prior to analysis by LC-MS.

LC-MS sample preparation. 1 mL of 2:1 chloroform:methanol was added to washed cell pellets, followed by 400 μL of acetone. The samples were vortexed, sonicated and lysed after adding a 5 mm stainless steel ball bearing (Qiagen, Manchester, UK) using a VelociRuptor tissue lyser (SLS, Wilford, Nottingham, UK). After centrifugation at $20,000 \times g$ for 5 min the supernatant was transferred to a separate glass vial and dried down using a centrifugal evaporator (Savant, Thermofisher, Horsham, UK). Dried samples were reconstituted in 200 μL of 2:1:1 2-propanol: acetonitrile: water for analysis.

LC-MS analysis of cholesterol. Lipid analysis for cholesterol was carried out using a Waters Premier CSH C18 column ($100 \times 2.1$ mm, 2.0 μm) using a ThermoFisher Vanquish Horizon LC system. Mobile phase A consisted of 6:4 acetonitrile: water with 10 mM ammonium formate and mobile phase B was 9:1 2-propanol: acetonitrile with 10 mM ammonium formate. For gradient elution of compounds mobile phase B started at 40% with an increase to 43% at 1.6 min, an increase to 50% at 1.7 min followed by a linear gradient to 54% B to 9.6 min, a further increase to 70% over 0.1 min followed by a linear gradient to 99% B over 4.7 min with re-equilibration for 1.5 min giving a total run time of 16 min. The flow rate was 0.5 ml/min and the injection volume was 5 μl. The needle wash used was 9:1 2-propanol: acetonitrile: water with 0.1% formic acid. The mass spectrometer used was a ThermoFisher Q Exative Orbitrap. Source parameters used for the mass spectrometer were a vaporiser temperature of 450 °C and ion transfer tube temperature of 320 °C, an ion spray voltage of 3.5 kV and a sheath gas of 55, auxiliary gas of 15 and a sweep gas of 3 arbitrary units with an S-lens RF (radio frequency) of 60%. For MS analysis a full scan of 120–1500 $m/z$ was used in positive ion mode at a resolution of 70,000 ppm. The mass resolution of cholesterol at 369 $m/z$ is calculated at 56823 FWHM. All solvents and additives used for all extractions and mobile phases were LC-MS or Optima grade and obtained from Fisher Scientific or Merck.

LC-MS data processing. All data were acquired using Xcalibur (Version 4.1, ThermoFisher Scientific). Targeted processing was carried out using Xcalibur. In order to confirm identification of compounds retention times were validated against known external standard solutions. Where appropriate peak areas corresponding to cholesterol levels were normalised to unlabelled cholesterol to account for varying amounts of material (Supplementary Data 3). Illustrative chromatograms and mass spectra are provided in Supplementary Data 4 and Supplementary Fig. 4.

## Statistical analyses

Quantification and data analysis of experiments are expressed as mean ± standard deviation and *P* values were calculated using analysis of variance (ANOVA) or two-tailed Student's t-test for pairwise comparisons, and were calculated using Graphpad Prism v.8. Qualitative experiments were repeated independently to confirm accuracy. Immunoblots were quantified using ImageJ software[53]. The mutagenesis screens were analysed using MAGeCK v.0.5.5[50], as described earlier.

## Data availability

All data reported in this paper will be shared by the lead contact upon request. The code for the CRISPR screen pipeline analysis can be found at 10.5281/zenodo.8135219. Source data are provided with this paper. In addition, source data for the HMGCR-Clover screen is included in Supplementary Data 1, the SREBP2-Clover screen in Supplementary Data 2, and the LC-MS cholesterol analysis in Supplementary Date 3 and 4. Any additional information required to reanalyse the data reported in this paper is available from the lead contact upon request. Source data are provided with this paper.

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

## Acknowledgements

We thank all members of the Nathan lab for their helpful comments on the manuscript. We also thank Richard Stopforth for help with the LC-MS analysis. This work was supported by a Wellcome Senior Clinical Research Fellowship to J.A.N. (215477/Z/19/Z), an MRC project grant (MR/X008118/1) to J.A.N., a Lister Institute Research Fellowship to J.A.N., an MRC PhD studentship to A.S.D., a Pfizer ITEN award to J.A.N., a Wellcome Trust Principal Research Fellowship to P.J.L. (210688/Z/18/Z), an MRC project grant (MR/V011561/1) to P.J.L., a Wellcome Investigator Award to A.K. (222497/Z/21/Z), a National Institutes of Health (NIH) grant R01CA201276 to D.V. This work was also supported by the NIHR BRC.

## Author contributions

Conceptualisation, A.S.D., T.P., B.M.O., P.J.L. and J.A.N.; Methodology, A.S.D., T.P., E.A., B.M.O., A.W.M., J.W., N.W., D.V., A.K., P.J.L. and J.A.N.; Investigation, A.S.D., T.P., E.A., B.M.O., A.W.M., J.W., N.V., Z.L., N.W., D.V., and J.A.N.; Writing – original draft, J.A.N.; Writing – reviewing and editing, all authors; Funding acquisition, J.A.N.; Resources, D.V., A.K., P.J.L. and J.A.N.; Supervision, J.A.N.

## Competing interests

J.A.N. receives a Pfizer ITEN discovery grant for unrelated work to this manuscript. Other authors declare no competing interests.
