## [Peer Review File · Nature Communications]

REVIEWER COMMENTS

Reviewer #1 (Remarks to the Author):

In this study, Dikson et al identify a HIF-independent mechanism for hypoxia changes in cholesterol synthesis. Using CRISPR screens the authors identify NADPH-MARCHF6 axis of controlling the transcription factor SREBP2. Overall this is a very interesting study, with the data supporting the authors conclusions and hypothesis. It is also significant since it has implications for sensitivity to statin treatment.

I just have some minor points relating to some of the experimental data provided.

First, although the genetic ablation of MARCHF6 greatly restores SREBP2 levels, it is never complete despite this being expected in a complete knockout. First are these absolute knockouts and if so, could the authors possibly discuss the action of a DUB?

Second, the authors show a gradient of hypoxia response but the difference between 21 and 5 is very big, so additional levels of oxygen would be needed to really analysis this.

Third, the authors should have provided NADPH/NAD measurement in 1% and not 0.5% since none of the experiments were done using this value of oxygen.

Finally, on page 8, line 224, the authors should change the word demonstratng to supporting the hypothesis/notion that NADPH is a ligand for MARCHF6. This is since the authors have not demonstrated this direct interaction in this work.

Are there any flaws in the data analysis, interpretation and conclusions? Do these prohibit publication or require revision?

Is the methodology sound? Does the work meet the expected standards in your field?

Is there enough detail provided in the methods for the work to be reproduced?

Reviewer #2 (Remarks to the Author):

In this manuscript, Dickson and colleagues identify a novel oxygen-sensing cellular mechanism: the hypoxia-induced degradation of SREBP2. Building upon the essential role of SREBP2 on sterol synthesis, the authors show that the hypoxia-NAPDH-MARCH6 axis decreases SREBP2 levels and inhibits cholesterol synthesis overriding the sterol-sensing pathways that usually govern SREBPs. The manuscript tests a very interesting hypothesis: because cholesterol synthesis is a highly oxygen-consuming process, there may be oxygen-sensing mechanisms rewiring this pathway. This has been previously observed in yeast, but this manuscript is the first one to formally test this idea in mammalian cells. Overall, the conclusions of the paper are well-supported by a variety of experimental models including fluorescent reporters, isotope tracing and functional genetic screens. However, I believe the addition of the following experiments would complement the current and strengthen the findings.

- In Extended fig 1A-B, the authors show full-length (FL) and cleaved SREBP2 levels in two additional cell lines (HepG2 and HEK293T). Although FL levels decrease under hypoxia, the levels of cleaved SREBP2 in these two cell lines are still quite high at 1% oxygen – unlike in HeLa cells (Fig. 1B). Because cleaved-SREBPs are the ones with transcriptional regulation potential, can the authors confirm that SREBP2 signaling is inhibited under hypoxia in these two additional cell lines?

- In Figure 2, western blot and fluorescent-reporter experiments show that HMGCR, the key enzyme initiating cholesterol synthesis, is degraded under hypoxia. In these first figures, the authors use what they call sterol depletion or StD (extract from Figure 1 legend: “DMEM supplemented with 10% lipid depleted FCS and 10 μ M mevastatin”). Because mevastatin binds and inhibits HMGCR, I would recommend to repeat some of these HMGCR stability experiments using a cholesterol synthesis inhibitor that doesn't inhibit HMGCR directly – for example the SQLE inhibitor NB-598 maleate.

- The CRISPR screens identifying MARCH6 as an essential player in SREBP2 degradation are very nice. However, for validation experiments the authors decide to use siRNAs. I would advise to pick the top sgRNA in the CRISPR screen and use it to generate MARCH6_KOs in a couple of cell lines to validate

its role in SREBP2 degradation.

- Is there a MARCH6 antibody available that could be used to complement the qPCR data of Fig. 4D?
- Because of the results on the CRISPR screen, I would expect MARCH6_KOs to have a significant effect on N-SREBP2 and HMGCR levels. The authors, however, only show the effect of MARCH6 and TRC8 double KOs. Could you add the single KOs experiments to define whether MARCH6 or TRC8 alone can trigger this regulation, or whether only one of them is the driver?
- All the experiments in the paper are done in vitro. But whether tumor hypoxia in vivo regulates SREBP2 and HMGCR is not tested throughout the paper. These are challenging experiments but for example pimonidazole staining coupled with immunofluorescence may be an option.

Reviewer #3 (Remarks to the Author):

This is certainly an interesting manuscript where the authors provide an explanation for changes in cholesterol synthesis according to oxygen levels. The authors explain a decrease in cholesterol synthesis in terms of MARCH6 mediated proteasomal degradation of SREBP2. While much of the manuscript is convincing there are a few matters that can be clarified.

Major Points

1. The manuscript suggests that cholesterol synthesis is curtailed in hypoxia as a result of enhanced degradation of SREBP2. A key experiment is [¹³C] enrichment to monitor new cholesterol production. The authors need to explain the mass spectrometry method used in more detail. What instrument? What resolution at m/z 400? What mass accuracy? How were cholesterol isotopomers normalised? Can you normalise a variable to a variable i.e. cholesterol? The authors need to provide an illustrative chromatogram and mass spectrum to allow the reader to assess the signal to noise ratios, resolution, mass accuracy and compare the levels of newly synthesised cholesterol to the unlabelled moiety.
2. Did the authors consider measuring the [¹³C] content of cholesterol esters? Action of SOAT is important in the homeostasis of cholesterol.
3. In a number of experiments n = 2. Is this usual practise in this field?
4. I was confused by the data with respect to HMGCR stabilisation. I was under the impression that HMGCR ubiquitination was mediated by INSIG and lanosterol (DOI 10.1016/j.cell.2005.12.022), so I am not so sure where SREBP2 degradation comes in here.

Other points.

5. In terms of sterol chemistry what is lipid-depleted serum? Different sterols have different activities towards SCAP, INSIG and HMGCR so the sterol composition is relevant to the interpretation of the experimental data.
6. Cholesterol is not a precursor of vitamin D. Vitamin D3 is derived from 7-dehydrocholesterol.
7. Perhaps some mention of LXR is required, particularly with respect to IDOL.

Response to Reviewers

We thank the Reviewers for their supportive and helpful comments. We believe the manuscript has been substantially improved by the additional work suggested. Our responses to the specific concerns are outlined, and we have also included the new/modified figures for ease of reference.

Reviewer #1 (Remarks to the Author):

In this study, Dikson et al identify a HIF-independent mechanism for hypoxia changes in cholesterol synthesis. Using CRISPR screens the authors identify NADPH-MARCHF6 axis of controlling the transcription factor SREBP2. Overall this is a very interesting study, with the data supporting the authors conclusions and hypothesis. It is also significant since it has implications for sensitivity to statin treatment.

We are pleased that this Reviewer found our work interesting and recognises the significant applications of the studies. We have addressed the minor points raised below.

I just have some minor points relating to some of the experimental data provided. First, although the genetic ablation of MARCHF6 greatly restores SREBP2 levels, it is never complete despite this being expected in a complete knockout. First are these absolute knockouts and if so, could the authors possibly discuss the action of a DUB?

The experiments detailed in Fig. 4e, f are complete knockouts for both MARCHF6 and TRC8. The MARCHF6 and TRC8 null CL1 cells have been verified in our prior work (PMID: 29519897). The rescue of FL SREBP2 is marked with MARCHF6 and TRC8 loss (Fig. 4f) but we agree with the Reviewer that N-SRE is not complete. It is possible that a further ligase controls just N-SRE stability in hypoxia and this could explain the remaining ubiquitination observed in the combined MARCHF6/TRC8 KO. It is also possible that a DUB is involved, but this has not yet been identified. We did examine the role of USP19, as it has been reported to control MARCHF6 stability (PMID: 25088257), but have not seen any involvement in our models. We now include the potential involvement of DUBs in the **Discussion**.

Second, the authors show a gradient of hypoxia response but the difference between 21 and 5 is very big, so additional levels of oxygen would be needed to really analysis this.

We agree that this is important, and now include further oxygen concentrations, at 12% and 7%, and see a graded response to oxygen availability (**New Fig. 3g, h; Supplementary Fig. 3j**). As expected, the changes are small between the oxygen ranges of 12% to 5%. It is also noteworthy that incubating cells in 12.5% oxygen does alter SREBP2 levels but is not sufficient to stabilise HIF-1 α (**New Supplementary Fig. 3j**), indicating that this SREBP2 oxygen-sensitive pathway is actually active across a broader range of oxygen tensions than the HIF pathway.

New Fig. 3g, h: N-SRE (g) and HMGCR levels (h) in HeLa control or HeLa HIF1 β KO cells cultured in 21%, 12%, 7%, 5% or 1% oxygen, with or without sterol depletion for 24 hr. Protein levels were quantified by ImageJ following normalisation to β -actin. **New Supplementary Fig 3j:** SREBP2 and HMGCR levels in HeLa control or HeLa HIF1 β KO cells cultured in 21%, 12%, or 7% oxygen, with or without StD for 24 hr. N-SRE Ct 21% versus 1% oxygen, $P < 0.0001$, N-SRE Ct 21% versus 1%, $P < 0.0001$, N-SRE HIF1 β KO 21% versus 1% oxygen, $P < 0.0001$, N-SRE HIF1 β KO 21% versus 1%, $P < 0.0001$, HMGCR Ct 21% versus 1% oxygen, $P < 0.0001$, HMGCR Ct 21% versus 1%, $P < 0.0001$, HMGCR HIF1 β KO 21% versus 1% oxygen, $P < 0.0001$, HMGCR HIF1 β KO 21% versus 1%, $P < 0.0001$. $n = 3$ biological repeats, mean \pm SD. $**P \leq 0.01$, $***P \leq 0.001$ Two-way ANOVA.

Third, the authors should have provided NADPH/NAD measurement in 1% and not 0.5% since none of the experiments were done using this value of oxygen.

These experiments in 0.5% oxygen were done alongside a prior study of lipid depletion in hypoxia (PMID 35739397). We thought it was helpful to include these to also allow comparison to prior work showing changing in hypoxia and lipid synthesis. However, we have now repeated these experiments in 1% oxygen using our sterol depletion, and not just de-lipidated media, with or without incubation in 1% oxygen (New Fig. 5e). This shows a similar perturbation in the NADPH/NADP⁺ ratio as was observed in 0.5% oxygen, and is more pronounced in sterol depletion. Please note that the values and ratios are different in the revised experiment due to a different NADPH assay set up.

New Fig. 5e: Relative NADPH/NADP⁺ levels in HeLa cells following incubation in 21% or 1% oxygen, with or without StD for 24 hr. Control 21% versus 1% oxygen, $P = 0.006$, StD 21% versus 1% oxygen, $P < 0.0001$. $n = 3$ biological repeats, mean \pm SD. $**P \leq 0.01$, $***P \leq 0.001$ Two-way ANOVA.

Finally, on page 8, line 224, the authors should change the word demonstratng to supporting the hypothesis/notion that NADPH is a ligand for MARCH6. This is since the authors have not demonstrated this direct interaction in this work.

This has been corrected.

Reviewer #2 (Remarks to the Author):

In this manuscript, Dickson and colleagues identify a novel oxygen-sensing cellular mechanism: the hypoxia-induced degradation of SREBP2. Building upon the essential role of SREBP2 on sterol synthesis, the authors show that the hypoxia-NADPH-MARCH6 axis decreases SREBP2 levels and inhibits cholesterol synthesis overriding the sterol-sensing pathways that usually govern SREBPs. The manuscript tests a very interesting hypothesis: because cholesterol synthesis is a highly oxygen-consuming process, there may be oxygen-sensing mechanisms rewiring this pathway. This has been previously observed in yeast, but this manuscript is the first one to formally test this idea in mammalian cells. Overall, the conclusions of the paper are well-supported by a variety of experimental models including fluorescent reporters, isotope tracing and functional genetic screens.

However, I believe the addition of the following experiments would complement the current and strengthen the findings.

We thank the Reviewer for the appreciative comments on our work and have addressed the additional points raised below.

- In Extended fig 1A-B, the authors show full-length (FL) and cleaved SREBP2 levels in two additional cell lines (HepG2 and HEK293T). Although FL levels decrease under hypoxia, the levels of cleaved SREBP2 in these two cell lines are still quite high at 1% oxygen – unlike in HeLa cells (Fig. 1B). Because cleaved-SREBPs are the ones with transcriptional regulation potential, can the authors confirm that SREBP2 signaling is inhibited under hypoxia in these two additional cell lines?

We thank the reviewer for raising these points. We now include qPCR data which confirms that SREBP2 transcriptional activity is reduced in the HepG2 cells and 293T cells to levels that correlate with N-SRE (New Supplementary Fig. 2c, d).

New Supplementary Fig. 2c, d: mRNA expression of SREBP2 target genes (*HMGCR* and *HMGCS1*), the SREBP1 and 2 target *LDLR*, and the HIF-1 target gene *CA9* in HepG2 (**c**) or HEK293T cells (**d**) treated with or without StD in 21% and 1% oxygen for 42 hr. $n=3$ biological repeats, mean \pm SD. * $P \leq 0.05$, *** $P \leq 0.001$ Two-way ANOVA.

- In Figure 2, western blot and fluorescent-reporter experiments show that HMGCR, the key enzyme initiating cholesterol synthesis, is degraded under hypoxia. In these first figures, the authors use what they call sterol depletion or StD (extract from Figure 1 legend: “DMEM supplemented with 10% lipid depleted FCS and 10 μ M mevastatin”). Because mevastatin binds and inhibits HMGCR, I would recommend to repeat some of these HMGCR stability experiments using a cholesterol synthesis inhibitor that doesn’t inhibit HMGCR directly – for example the SQLE inhibitor NB-598 maleate.

This is an important point and we now include the HMGCR stabilisation assay with NB-598. Combined lipid depletion with NB-598 stabilised our HMGCR-Clover reporter in 21% oxygen, in a similar manner to lipid depletion plus mevastatin (New Supplementary Fig. 2e). Importantly, HMGCR levels are still suppressed following lipid depletion and NB-598 treatment in hypoxia (New Supplementary Fig. 2e). Therefore, our findings are not specific to statins. We think it is important to highlight that we and others have previously used combined 10% lipid depleted media and mevastatin treatment to interrogate the degradation of HMGCR, and we believe that it is the appropriate system to use in our assays.

New Supplementary Fig. 3a: Flow cytometry analysis of HMGCR-Clover levels following incubation in 21% or 1% oxygen, with or without 10% lipid depleted media (LD) plus the SQLE inhibitor NB-598 maleate (1 μ M) for 24 hr. *Representative of 3 independent experiments.*

- The CRISPR screens identifying MARCH6 as an essential player in SREBP2 degradation are very nice. However, for validation experiments the authors decide to use siRNAs. I would advise to pick the top sgRNA in the CRISPR screen and use it to generate MARCH6_KOs in a couple of cell lines to validate its role in SREBP2 degradation.
- Is there a MARCH6 antibody available that could be used to complement the qPCR data of Fig. 4D?

We thank the reviewer for their appreciation of our screening approach. Unfortunately, there are no suitable antibodies to detect MARCHF6. We have tried many antibodies over the years and none of them work. This is a recognised problem for examining MARCHF6 and no group that we are aware of, or have contacted, have been able to examine MARCHF6 endogenously.

Our usual approach for validating screens is to do exactly as this reviewer suggests - take several sgRNA to generate mixed KO populations and clones (e.g. PMID: 34155378, PMID: 32792488), and we did this in our prior work examining a role of TRC8 and MARCHF6 in the degradation of misfolded proteins (PMID: 29519897). These clones (including their generation from the mixed KO populations, clonal validation, and reconstitutions) have already been described in published work (PMID: 29519897). However, we agree that it is helpful to include the data for the single MARCHF6 and TRC8 KO clones, which confirm that loss of MARCHF6 increases SREBP2 levels (**New Supplementary Fig. 5b**), consistent with our findings with siRNA targeting MARCHF6 and the combined TRC8/MARCHF6 null clones.

We would like to highlight that siRNA-mediated MARCHF6 depletion provides an orthogonal approach to sgRNA used in the screen. SiRNA also allowed us to accurately quantify the efficiency of the knockdown of MARCHF6, which is important given the lack of a suitable antibody.

- Because of the results on the CRISPR screen, I would expect MARCH6_KOs to have a significant effect on N-SREBP2 and HMGR levels. The authors, however, only show the effect of MARCH6 and TRC8 double KOs. Could you add the single KOs experiments to define whether MARCH6 or TRC8 alone can trigger this regulation, or whether only one of them is the driver?

It is interesting that the screen only identified MARCHF6, whereas our prior proteomic analysis identified both MARCHF6 and TRC8 as potentially involved in the degradation of SREBP2 (PMID: 29519897). The siRNA experiments targeting MARCHF6 or TRC8 suggested that MARCHF6 was the dominant ligase for SREBP2 stability. We have now included experiments with the single MARCHF6 and TRC8 null clones in hypoxia, with or without sterol depletion (**New Supplementary Fig. 5b, c**). N-SRE and HMGR are stabilised to a greater level in the MARCHF6 compared to the TRC8 null clones in combined hypoxia and sterol depletion, but both show an effect. These findings are consistent with the siRNA experiments, whereby MARCHF6 seems to be the dominant ligase. We postulate that MARCHF6 is the main ligase involved in the hypoxic regulation of SREBP2, whereas TRC8 may be involved in its basal regulation. We plan to study this further in our subsequent work, and whether TRC8 has a NADPH dependency, similarly to MARCHF6.

New Supplementary Fig. 5b, c: (b, c) HeLa mCherry-CL1 control (Ct), MARCHF6 (b) or TRC8 (c) HeLa mCherry-CL1 clonal null cells were treated with StD for 42 hr, with or without 1% oxygen for the final 18 hr. SREBP2 processing and HMGCR levels were analysed by immunoblot. *Representative of 3 independent experiments.*

- All the experiments in the paper are done in vitro. But whether tumor hypoxia in vivo regulates SREBP2 and HMGCR is not tested throughout the paper. These are challenging experiments but for example pimonidazole staining coupled with immunofluorescence may be an option.

These types of experiments are of significant interest to us, and the focus of our future work. Our current studies in this area are preliminary and we believe they are beyond the scope of this paper, where our focus has been to uncover the mechanistic basis for the HIF independent regulation of cholesterol synthesis.

For the benefit of this Reviewer, we include below some preliminary data where we have undertaken some immunofluorescence imaging studies using a version of bodipy for sterols (TNM-BF, PMID: 20543850, 24386262), which stains sterol-containing membranes in cells. TNM-BF levels are reduced following sterol depletion (statin and lipid depletion) or statin treatment alone. Hypoxia markedly reduces TNM-BF staining compared to 21% oxygen, particularly when combined with sterol depletion. DMOG treatment, unlike hypoxia, does not reduce TNM-BF levels, consistent with hypoxic regulation of cholesterol synthesis being independent of the HIF response (**Rebuttal Fig. 1**).

As this reviewer points out, these experiments are challenging, and immunofluorescence for SREBP2 or HMGCR has not been successful. In addition, pimonidazole will not distinguish between perturbations in hypoxia or the NADPH/NADP redox state in tissues, as both will affect adduct formation with the compound. We therefore do not think it is appropriate to include these preliminary studies in this manuscript, but we plan to address these issues using our HIF dynamic reporters alongside other hypoxia-probes in mouse tumour models in future work.

Rebuttal Fig. 1: Immunofluorescence of HeLa cells with TNM-BF to stain for cholesterol. Cells were seeded and treated with or without sterol depletion (StD), in 21% or 1% oxygen, or with DMOG as indicated for 24 hr. Cells were fixed with 4% PFA, treated with 0.1% saponin, prior to staining with TNM-BF (1 μ g/ml). Calnexin was used to stain for the ER.

Reviewer #3 (Remarks to the Author):

This is certainly an interesting manuscript where the authors provide an explanation for changes in cholesterol synthesis according to oxygen levels. The authors explain a decrease in cholesterol synthesis in terms of MARCHF6 mediated proteasomal degradation of SREBP2. While much of the manuscript is convincing there are a few matters that can be clarified.

We thank the Reviewer for their appreciation of our work and their helpful comments. We have addressed the points below.

Major Points

1. The manuscript suggests that cholesterol synthesis is curtailed in hypoxia as a result of enhanced degradation of SREBP2. A key experiment is [13 C] enrichment to monitor new cholesterol production. The authors need to explain the mass spectrometry method used in more detail. What instrument? What resolution at m/z 400? What mass accuracy? How were cholesterol isotopomers normalised? Can you normalise a variable to a variable i.e. cholesterol? The authors need to provide an illustrative chromatogram and mass spectrum to allow the reader to assess the signal to noise ratios, resolution, mass accuracy and compare the levels of newly synthesised cholesterol to the unlabelled moiety.

We apologise for these methodological omissions. These have now been included. Cholesterol isotopomers were normalised to the unlabelled cholesterol in the relevant sample, which is our and others' standard practice when processing uptake of labelled substrate in LC-MS data (e.g. PMID 31978345). The theory being that for samples that have varying amounts of material (i.e. cell number) by dividing the signal from one isotopomer by another the contribution to the signal from differing total amount of material 'cancels out'. This is not tantamount to dividing one variable by another, as the variables are not independent. In our opinion, the most appropriate way to interpret this type of data is by showing fractional incorporation. Other methods of accounting for the differing

total signal from sample to sample such as normalising to total protein or DNA content needlessly add an extra source of variability by including an extra assay.

Illustrative mass spectra and a chromatogram are now included for a control sample and cells treated with lipid depletion (**Supplementary Data 4** and **New Supplementary Fig. 4**). The M+0 unlabelled cholesterol is observed in both conditions, but the various labelled [¹³C]cholesterol isotopomers are only observed in lipid depleted conditions, consistent with cholesterol synthesis. The M+13 and M+0 chromatogram illustrates the m/z and mass accuracy for the isotopomers.

New Supplementary Fig. 4: (a, b) Illustrative mass spectra for HeLa cells treated with (a, top panel) or without lipid depletion (a, bottom panel), and a chromatogram of M+0 and M+13 from the lipid depleted sample (b). The raw data is included in **Supplementary Data 4**.

2. Did the authors consider measuring the [13C] content of cholesterol esters? Action of SOAT is important in the homeostasis of cholesterol.

We did attempt to measure cholesterol esters but these were not well detected in our LC-MS system. Unfortunately, we only have access to an electrospray ion source coupled to a Q Exactive MS. Cholesterol esters ionise very poorly during electrospray ionisation and all that can be detected are in source fragments corresponding to the cholesterol head group. They can, however, be routinely measured using APCI (atmospheric pressure chemical ionisation) but unfortunately we do not have access to this type of source.

We acknowledge that SOATs are important for cholesterol esters, but they are not required for the cholesterol sensing of SREBP2 and HMGCR at the ER membrane. Cholesterol ester production and their storage pool are likely altered in hypoxia. We know that HIF-2 can promote lipid droplet storage via PLIN2 (PMID 25829424), and our data indicates that cells become reliant on the uptake of cholesterol esters in hypoxia, as SREBP2 activity is impaired. Whether hypoxia directly regulate SOATs is an interesting question, and an area we would like to address in our future studies. We do agree that it is helpful to discuss cholesterol ester formation and storage in relation to our findings (please see revised **Discussion**).

3. In a number of experiments n = 2. Is this usual practise in this field?

We agree that it is helpful to have three biological replicates where possible. However, two biological replicates were used in the [13C] tracing experiments in view of the time, cost and analysis of the metabolic tracing. We would also highlight that the orthogonal approaches clearly show that cholesterol synthesis is perturbed, as HMGCR is the most sensitive readout for the initiation of cholesterol synthesis. At least 3 biological replicates are included for all other experiments, aside from a couple (e.g. Supplementary Fig 2c,d (HMGCR levels in HepG2 and 293T cells) and Supplementary Fig 1c,d (SREBP1 levels) where images are representative of 2 experiments. However, these experiments have been done in multiple cell lines (HeLa, HepG2, 293T, 786-O) and orthogonal approaches have confirmed the same findings.

4. I was confused by the data with respect to HMGCR stabilisation. I was under the impression that HMGCR ubiquitination was mediated by INSIG and lanosterol (DOI 10.1016/j.cell.2005.12.022), so I am not so sure where SREBP2 degradation comes in here.

We apologise that this was not made clear. HMGCR levels are controlled by two mechanisms: (1) HMGCR transcriptional regulation under the control of SREBP2, and (2) ubiquitin-mediated degradation involving INSIGs. SREBP2 mediated transcription of HMGCR is equally as important as the post-translational regulation of HMGCR, with loss of SREBP2 preventing cholesterol synthesis via reduced HMGCR transcription and a reduction in other cholesterol synthetic genes (e.g. PMID: 9150132, Brown and Goldstein). We can demonstrate the importance of the SREBP2 arm of HMGCR regulation using the HMGCR-Clover reporter cells, where sgRNA depletion of SREBP2 prevents the accumulation of HMGCR-Clover following sterol depletion (**New Supplementary Fig. 2**).

New Supplementary Fig. 2e: HeLa HMGCR-clover or mixed KO populations of SREBP2 were treated with sterol depletion for 42 hr, with or without incubation in 1% oxygen for the final 18 hr. Cells were then analysed by live cell flow cytometry.

Other points.

5. In terms of sterol chemistry what is lipid-depleted serum? Different sterols have different activities

towards SCAP, INSIG and HMGCR so the sterol composition is relevant to the interpretation of the experimental data.

The lipid depleted media was batch purchased from Biosera. Lipids are removed using a fumed silica precipitation. The acceptable level of lipid depletion is cholesterol lower than 10 mg/100ml, according to the manufacturer. The silica precipitation method is well established for all industry related lipid-depleted media, and non-polar substances such as lipids and steroids will adsorb to the surface of fumed silica. There is therefore no selective removal of just cholesterol. This is also evident by our control experiments that show that SREBP2 is cleaved in the lipid depleted media, and that HMGCR is stabilised.

6. Cholesterol is not a precursor of vitamin D. Vitamin D3 is derived from 7-dehydrocholesterol.

We have corrected this in the text to highlight Vitamin D3 is derived from the cholesterol synthetic pathway.

7. Perhaps some mention of LXR is required, particularly with respect to IDOL.

We agree, and have included this in the Discussion, as to our knowledge it is not yet known how hypoxia or HIFs regulate the LXR-Idol-LDL receptor axis.

REVIEWER COMMENTS

Reviewer #1 (Remarks to the Author):

The authors have done a very effort in addressing all the reviewers comments and the manuscript is now even stronger

Reviewer #2 (Remarks to the Author):

All my comments and concerns were addressed. The latest version of the manuscript properly supports the author's conclusions.

Reviewer #3 (Remarks to the Author):

The authors do provide substantial amount of data to support their conclusions that SREBP2 acts as an oxygen-sensitive regulator of cholesterol synthesis which is in the most part convincing.

However, I am not sure that an n of only 2 is acceptable in Figure 2g and 2h. Neither am I convinced that cholesterol isotopomers can be normalized against cholesterol, as to my mind in the experiments conducted cholesterol is a variable. Perhaps to clear this point the authors could re-label the y-axis as a ratio and allow the reader to make up their own mind.

Supplemental Figure 4 does not look like LC-MS data from an Orbitrap. In my experience there should be evidence of chemical noise, this appears to be absent.

Other points

1. Line 55, the statement "When cholesterol is present, SREBP2 ---" could be rephrased. See doi: 10.1016/j.cmet.2008.10.008.
2. Line 556, [13C] not [C13].
3. Line 582, what is the mass resolution at m/z 400?
4. Supplemental Fig 4, 369 is not M of cholesterol it is [M+H-H₂O]⁺.

Response to Reviewers

Reviewer #1 (Remarks to the Author):

The authors have done a very effort in addressing all the reviewers comments and the manuscript is now even stronger

Reviewer #2 (Remarks to the Author):

All my comments and concerns were addressed. The latest version of the manuscript properly supports the author's conclusions.

We thank both Reviewers for their appreciation of our work.

Reviewer #3 (Remarks to the Author):

The authors do provide substantial amount of data to support their conclusions that SREBP2 acts as an oxygen-sensitive regulator of cholesterol synthesis which is in the most part convincing.

However, I am not sure that an n of only 2 is acceptable in Figure 2g and 2h. Neither am I convinced that cholesterol isotopomers can be normalized against cholesterol, as to my mind in the experiments conducted cholesterol is a variable. Perhaps to clear this point the authors could re-label the y-axis as a ratio and allow the reader to make up their own mind.

We appreciate the recognition of our work. We continue to disagree regarding the normalisation. However, irrespective of this, displaying the data normalised to cholesterol or not does not alter the findings. We have therefore just displayed the cholesterol isotopomer levels not normalised to cholesterol (**New Figure 2h**) to address the concerns raised.

We again point the other substantial amount of data and orthogonal approaches that clearly show that cholesterol synthesis is perturbed in hypoxia.

Supplemental Figure 4 does not look like LC-MS data from an Orbitrap. In my experience there should be evidence of chemical noise, this appears to be absent.

The LC-MS data is from an Orbitrap. The LC-MS extracted data was included in **Supplemental Data 4** (Excel File). A screenshot of non-extracted chromatogram LC-MS Orbitrap source data is shown below.

Other points

1. Line 55, the statement "When cholesterol is present, SREBP2 ---" could be rephrased. See doi: 10.1016/j.cmet.2008.10.008.

We are not sure what the concern is here but we have clarified that this relates to the relative level of cholesterol.

2. Line 556, [13C] not [C13].

This has been corrected.

3. Line 582, what is the mass resolution at m/z 400?

The full scan of 120-1500 m/z was used in positive ion mode as stated at a resolution of 70,000ppm.

The mass resolution of cholesterol at m/z 369 is calculated at 56,823 FWHM.

4. Supplemental Fig 4, 369 is not M of cholesterol it is [M+H-H₂O]⁺.

This has been corrected.